

# Influence of extreme long-term rainfall and unsaturated soil properties on triggering of a landslide - a case study

Håkon Heyerdahl[1, 2]

[1]Department of geosciences, Faculty of Mathematics and Natural Sciences, University of Oslo (UiO), Oslo, 0316, Norway
[2]Norwegian Geotechnical Institute (NGI), Oslo, 0806, Norway

*Correspondence to*: Håkon Heyerdahl (hhe@ngi.no)

## Abstract

A case study of a natural slope in Eastern Norway that failed after extreme long-term rainfall in year 2000 was performed. Effect of soil suction on soil shear strength was investigated for intact specimens of Quaternary silt/sand, using an unsaturated

shear box apparatus. Common prediction models under-predicted the unsaturated shear strength, particularly for small suctions. Analyses of rainfall infiltration were performed for silt and sand slopes, based on retention curves measured in laboratory. For normal annual rainfall of 800 mm/year the model slope is theoretically stable. Extreme rainfall (240 mm in 30 days) during the autumn of year 2000 results in a rise of groundwater and loss of soil suction in the vadose zone. To reach theoretical slope failure, lower cohesion had to be assumed than measured in laboratory. High cohesion may be caused by

cementation in shallow soil layers, and lower cohesion may be appropriate. Slope stability analyses based on transient seepage analysis of rainfall show gradual decrease of slope stability towards slope failure (for the silt slope). With expected future increase in rainfall, more attention is needed on the role of unsaturated soil properties in rainfall triggering of landslides in different soil types, climatic conditions and geologic settings.

## 1 Introduction

Rainfall triggered landslides in intermediate and fine-grained soils occur regularly all over Norway in areas dominated by Quaternary sediments, e.g. Sandersen et al. (1996), Jaedicke and Kleven (2007), Heyerdahl (2016). The landslide activity is expected to increase as a result of predicted future climatic change (Jaedicke et al., 2008). Slope stability may rely on "apparent cohesion" resulting from negative pore-water pressures in the soil volume above the ground water line, i.e. the unsaturated, or vadose, zone. Evaluation of slope stability of natural slopes therefore should involve unsaturated soil mechanics (Fredlund and

Rahardjo, 1993). Internationally, studies of unsaturated soils have advanced immensely through the last decades, and applied in work with rainfall triggered landslides for many different climatic and geological conditions, e.g. Lim et al. (1996), Ng and Shi (1998), Tsaparas et al. (2000), Gasmo et al. (2000), Rahardjo et al. (2001), Collins and Znidarcic (2004), Farooq et al. (2004), Casagli et al. (2006), Zhan et al. (2007), Gavin and Xue (2008), Thielen and Springman (2008), Valentino et al. (2011), Zhang et al. (2011). On the other hand, the influence of soil suction and unsaturated soil parameters on slope stability has been





little investigated for Norwegian natural soils. In geotechnical practice in Norway, the effect of negative pore-water pressures on the soil's shear strength is, in fact, generally neglected, i.e. the soil suction is set equal to zero above groundwater level. Hence, many natural slopes may appear theoretically instable, and effect of rainfall and infiltration will only be taken into account through rise in groundwater level and increase in pore-water pressures. This conservative approach may be sound for

geotechnical design (since soil suction may dissipate), but offers no solution for improved understanding of landslide release processes, where infiltration processes in shallow soil layers may be decisive for landslide triggering. Unsaturated soil mechanics has been applied only in a few studies, and generally with little or no experimental data for unsaturated soil parameters available: Edgers and Nadim (2004) performed infiltration analyses for an unsaturated soil column to explain triggering of debris flows in Western Norway; Höydal and Heyerdahl (2006) discussed effect of long-term infiltration on safety

factor of clay slopes in Eastern Norway; l´Heureux et al. (2006) studied the failure of a natural sand slope, while Heyerdahl et al. (2013) applied unsaturated soil mechanics to explain failure of an old railway embankment constructed of clay, silt and sand.

The motivation for the work presented herein is therefore to fill some gaps when it comes to unsaturated soil behavior for Norwegian soils, in particular unsaturated shear strength and water retention properties of natural soils, and to study the

importance of these parameters for rainfall triggering mechanisms of landslides in Norway. The study is concentrated around a landslide site at the farm Negarden Sander in the municipality of Eidsvoll, Upper Romerike region in Eastern Norway. The landslide was one of many slopes failing in year 2000, following extreme long-term rainfall (Jaedicke and Kleven, 2007). In year 2000 at least four different shallow landslides occurred within a limited area (Fig. 1a). One of the slides occurred very close to a residential building (Fig. 1b), which had to be temporarily evacuated until the slope had been repaired. From

interpretation of a rotary sounding at the top of the slope, the ground conditions may be described as 6-7 m of sand/silt on top of marine clay down to 30 m depth (Fig. 1c).

Intact samples for laboratory testing were collected in a manually excavated sample pit (Fig. 2), positioned as indicated in Fig. 1a. Retention curves were measured for intact specimens of silt and sand from the site (Heyerdahl and Pabst, 2017), and some unsaturated shear box tests on intact silty sand specimens were performed (Heyerdahl, 2016). A more comprehensive series

of unsaturated shear box tests on specimens of a sandy silt from the test site is presented herein, together with numerical analyses of rainfall triggering based on the experimental results. In this study, exclusively intact soil specimens were used, in order to preserve the soil's in situ characteristics as much as possible. The advantage of intact specimens is that the in situ soil structure and porosity is preserved, in contrast to much basic research on unsaturated soils, which has been done on compacted specimens, e.g. Donald (1956); Bishop et al. (1960), Bishop and Blight (1963), Blight (1967), Ahmed et al. (1974), Fredlund

and Morgenstern (1977), Escario (1980), Alonso et al. (1990), Vanapalli et al. (1996a), Romero et al. (1999), Ng and Chiu (2001), Springman et al. (2003), Estabragh (2012). However, also the use of compacted sediments may have advantages, e.g. when it comes to the possibility of repeating tests with identical specimens, each having equal grain size distribution, density, void ratio etc.





## 2 Shear strength of unsaturated soil

### 2.1 Stress in unsaturated soil

For landslides triggered by infiltration of water, the unsaturated shear strength needs to be evaluated. Unsaturated shear strength of soils has been a topic of discussion since the 1950's , and still remains a question permanently subjected to research (Lu and

5 Likos, 2004). Bishop (1959) suggested a single-valued effective stress formulation for unsaturated soils, covering the complete range of water content from completely dry soil to completely saturated soil:

$$\sigma' = (\sigma - u_a) + \chi (u_a - u_w) \tag{1}$$

in which $u_a$ is air pressure, $u_w$ is pore-water pressure, $(u_a - u_w)$ is matric suction, $\sigma$ is total stress, $\sigma'$ is effective stress and $\chi$ is a material parameter. Bishop et al. (1960) combined the single-valued "unsaturated effective stress" from Eq. (1) with the

10 classical Mohr-Coulomb model for drained shear strength of soils, and suggested the following equation for the shear strength of unsaturated, cohesive soil:

$$\tau_f = c' + [\sigma - u_a + \chi(u_a - u_w)]\tan\varphi' \tag{2}$$

in which $\tau_f$ is the shear stress at failure, $c'$ is effective cohesion and $\phi'$ is the effective friction angle. From Eq. (2) the value of $\chi$ may be calculated from suction controlled shear tests, given that the remaining parameters are known. The parameter $\chi$ has

15 been found to vary with saturation rate, generally with a non-linear variation, e.g. Bishop et al. (1960), Bishop and Blight (1963).

Fredlund and Morgenstern (1977) showed that a single-valued stress state variable (analogous to Terzaghi's effective stress principle) generally is insufficient to describe mechanical behaviour of unsaturated soils, and that two independent stress state variables are necessary.

### 20 2.2 Shear strength prediction models for unsaturated soil

De-coupling the shear strength formulation into independent contributions from each of the stress state variables $(\sigma - u_a)$ and $(u_a - u_w)$, Fredlund et al. (1978) suggested that shear strength of unsaturated soils be modelled as a bi-linear function in an "extended Mohr-Coulomb function":

$$\tau = c + (\sigma - u_a)\tan\phi' + (u_a - u_w)\tan\phi^b \tag{3}$$

in which $\phi^b$ is the friction angle related to a change in matric suction. Two different friction angles hence were proposed for increase in vertical net stress and matric suction, respectively. For positive pore-water pressures, the classical Mohr-Coulomb strength formulation for saturated soil in Eq. (8) is reproduced. For negative pore-water pressures, the shear strength increases according to the product of suction and $\phi^b$, suggested to be around half the effective friction angle (Fredlund and Rahardjo, 1993). Experimental data however showed non-linearity of the shear strength curve along the matric suction axis for many

soils, e.g. Escario (1980), Escario and Saez (1986), and $\phi^b$ should not be considered a constant. Gan et al. (1988) discussed





non-linearity of the strength envelope of a glacial till, and concluded that $\phi^b$ is not necessarily constant for varying suction, but decreases from a value close to $\phi'$ for saturated soil to a relatively constant value for increased suction. To handle non-linearity, it was suggested to e.g. discretize the curve into linear segments with constant $\phi^b$. Although the bilinear model with constant $\phi^b$ is now generally rejected, the slope angle of shear strength versus suction (i.e. tangent) is still commonly called $\phi^b$. The

bilinear model may be applied within a limited suction range for which $\phi^b$ is assumed constant (which should be verified empirically). Below air entry value the soil is practically saturated, and $\phi^b$ should be equal to $\phi'$, e.g. Fredlund et al. (1987), Gan et al. (1988), Vanapalli et al. (1996b), Ferrari (2007). When pore-water gradually evacuates from the sample with increased suction, $\phi^b$ will start to drop. The principal of two independent stress variables (Fredlund and Morgenstern, 1977) has been applied in several other prediction models for unsaturated shear strength of soil. The main difference between the

models is the formulation of the parameter $\chi$. Vanapalli et al. (1996b) suggested two prediction models, both taking the water retention curve directly into account when predicting the unsaturated shear strength:

"Vanapalli's 1$^{st}$ approach":

$$\tau = c' + (\sigma_n - u_a)\tan\varphi' + (u_a - u_w)\Theta^\kappa \tan\varphi' \qquad (4)$$

in which $(\sigma_n - u_a)$ is the net normal stress on the failure plane at failure, normalized water content $\Theta = \theta / \theta_s$ where $\Theta$ is

volumetric water content, $\Theta_s$ is saturated volumetric water content, $\kappa$ is an exponent accounting for the relation between normalized water content and normalised area of water in the pores, $a_w$, and $a_w = \Theta^\kappa$. Relating to Eq. (2) the term $\Theta^\kappa$ is equivalent to the parameter $\chi$ and directly related to the water retention curve of the soil, with the parameter $\kappa$ as a fitting parameter. Good correlation was found between experimental results and predicted shear strength for $\kappa$ equal to 2.2 for a compacted glacial till (Vanapalli et al., 1996b).

"Vanapalli's 2$^{nd}$ approach":

$$\tau = c' + (\sigma_n - u_a)\tan\varphi' + (u_a - u_w)\tan\varphi'\left(\frac{\theta - \theta_r}{\theta_s - \theta_r}\right) \qquad (5)$$

in which $\Theta_r$ is residual volumetric water content. In Eq. (5), matric suction contributes to the shear strength proportionally with the product of matric suction and effective saturation rate. Again relating to Eq. (2), the term $(\theta - \theta_r)/(\theta_s - \theta_r)$ is equivalent to $\chi$ and directly given by the soil's water retention curve. Öberg and Sällfors (1997) suggested a similar formulation as the "2$^{nd}$

approach" of Vanapalli et al. (1996b). Considering the hypotheses presented above, increased shear strength from an increase in matric suction requires an increase in the products $(u_a - u_w)\Theta^\kappa$ or $(u_a - u_w)(\theta - \theta_r)/(\theta_s - \theta_r)$, respectively. Hence, the shear strength may increase, remain constant, or drop in the residual stage of desaturation (Vanapalli et al., 1996b), which is governed by the shape of the retention curve.

Khalili and Khabbaz (1998) suggested to predict undrained shear strength based on Bishop's effective stress with $\chi$ given by

Eq. (6) and (7):



$$\chi = 1 \, , \, u_a\text{-}u_w \leq AEV \tag{6}$$

$$\chi = \left[\frac{u_a - u_w}{AEV}\right]^{-0.55} , \, u_a - u_f > AEV \tag{7}$$

in which $AEV$ is the air entry value. For $u_a$ - $u_w \leq AEV$ the effective stress parameter $\chi$ is equal to 1, and Eq. (1) collapses to the classical Mohr-Coulomb shear strength expression in Eq. (8), in which $\sigma' = \sigma - u$ is Terzaghi's effective stress. However, since
pore-water pressure will be negative, $u_w$ contributes to *increased* shear strength.

$$(8) \qquad \tau_f = c' + (\sigma_v - u_w)\tan\phi$$

Kim and Borden (2011) compared prediction models for unsaturated shear strength for matric suctions < 200 kPa with test data from literature, and concluded that prediction methods similar to the models suggested by e.g. Vanapalli et al. (1996b) gave poor prediction of unsaturated shear strength for soils that desaturate fast. Somewhat better prediction was found using
the model suggested by Khalili and Khabbaz (1998). The prediction models by Vanapalli et al. (1996b) and Khalili and Khabbaz (1998) are compared with new test data later in this paper.

**3 Unsaturated shear tests**

Multi-stage direct shear tests were performed in the geotechnical laboratory of UPC, Barcelona, using an unsaturated shear box apparatus similar to the equipment described in Escario (1980).

**3.1 Multi-stage shear tests**

Multi-stage shear tests are commonly used for unsaturated soil testing (Fredlund and Rahardjo, 1993), because the amount of work and time needed for testing will be greatly reduced. Gan and Fredlund (1988) recommended to avoid passing the peak shear strength in each load stage to avoid structural breakdown. Reduction of accumulated strain through unloading between load stages was suggested. The authors pointed out the greatly time-saving effect of reduced specimen height in direct shear
box tests compared to triaxial tests, also emphasized by Escario (1980) and Escario and Saez (1986). Studies by Ng et al. (2007) confirmed that shear tests from single-stage and multi-stage tests gave comparable results. Normal testing procedure is to start at low vertical net stress or suction (and hence low shear strength) and increase the load stepwise in subsequent load stages (towards higher shear strength). To investigate suction effect in a multi-stage shear test, suction must consequently increase in each new load step. The soil will then follow the main drying branch of the retention curve. During rainfall, matric
suction will however drop due to infiltration, and the soil will follow the wetting branch of the retention curve. A favourable testing scheme for studies of rainfall-induced landslides would be to start from low water content and reduce suction gradually by allowing water to flow into the specimen, as in single-stage shear tests at constant vertical stress and increasing water content performed by Melinda et al. (2004), simulating load conditions during rainfall. Multi-stage testing following a load path towards lower matric suction is however not recommendable, since specimens may be damaged by preceding load stages.





The tests reported herein were executed as multi-stage tests, increasing either matric suction or vertical net stress in each new load step. The advantage is that more information is achieved from each specimen, and that results from different load stages are directly comparable, without the inherent variation that may exist between different specimens. The disadvantage of the testing procedure, as explained above, is that the load path is different from what will occur in situ under a rainstorm.

### 3.2 Sample collection and preparation of specimens

An intact brass cylinder sample (length 25 cm, diameter Ø=73 mm) was collected in a 1.4 m deep test pit at Eidsvoll, Norway (Fig. 1, Fig. 2), and brought to the geotechnical laboratory at UPC, Barcelona, for testing. In situ soil suction of approx. 25 kPa was measured in the silt layer with tensiometers at the time of sampling (Heyerdahl, 2016). Specimens were cut from the intact sample to fit into the 50 x 50 mm square shear box (Fig 3a). The top half of the shear box was slid down onto the sample before carefully releasing the specimen with a thread saw (Fig 3b). Specimens had horizontal layering, with occasional thin layers or pockets of fine sand.

### 3.3 Performing shear tests and controlling suction

Only one-directional displacement was possible; i.e. reverse displacement/unloading was done manually (tests *Silt S7* and *Silt S8*). Horizontal displacement was calculated from elapsed time and displacement rate. Horizontal load was measured by a 5 kN load cell. Vertical deformation was measured by a LVDT sensor at the vertical load piston. Sensors were logged at 0.1 Hz. Horizontal load was zeroed displacing the test cell at constant rate. Signal noise was reduced by floating averaging of readings. Water pressure in the water compartment below the porous disk (high air entry value disk) was controlled by a GDS pump. Five multi-stage tests on silt, tests *Silt S4* to *Silt S8*, were performed, each consisting of three to seven load steps. The lowest available displacement rate (0.005 mm/min) was used to ensure drained conditions (0.010 mm/min in test *Silt S4*). Vertical load was applied after closing the test chamber. Suction was controlled by the axis translation technique in most load steps, in order to keep pore-water pressure positive (Hilf, 1956). Suction is defined by the difference between air pressure in the lower test chamber and water pressure in the water compartment below the porous disk. Axis translation prevents cavitation and reduces air diffusion, normally limited to matric suctions up to 1500 kPa (Fredlund and Rahardjo, 1993). Stress state variables suggested by Fredlund and Morgenstern (1977) are not altered by axis translation. Results from parallel shear tests with the axis translation and osmotic techniques, respectively, verified that these methods produced similar results, with no significant difference for the effective friction angle (Ng et al., 2007). Slightly larger $\phi^b$ found for the axis translation technique was attributed to variations in specific volume.

For compaction at low saturation rates, the macro-pore volume of specimens may be reduced without much water leaving or entering. Vertical compression from application of vertical net stress hence may stabilize quickly (within few minutes). For changes in matric suction, longer consolidation time was needed. For tests *Silt S7* and *Silt S8*, overnight consolidation was used for load steps involving increased matric suction. Volume change versus time in the GDS pump was then observed to be linear, representing only air diffusion (Heyerdahl, 2016).

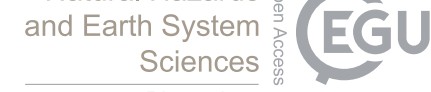



Three of the tests (*Silt S4*, *Silt S5* and *Silt S6*) started from in situ water content. In tests *Silt S7* and *Silt S8* specimens were initially saturated by filling water in the test chamber outside the shear box, letting the specimens saturate from below. Initial saturation of the specimen secures establishment of a continuous water phase between the porous disk and the specimen. Observed volume change of specimens during saturation was negligible. In the first load step these saturated specimens were

sheared with some free water in the lower test chamber to maintain full saturation during the test. After the first shearing, the specimens were unloaded by moving the test chamber away from the horizontal load piston, and free water was removed from the test chamber. Track of horizontal displacement is thereby lost. For evaluation of horizontal displacements throughout the test, horizontal load from the first load step was used as reference, assuming elastic unloading/reloading (indicated in Figs. 4 and Fig. 5). Matric suction in the second and third load steps of tests *Silt S7* and *Silt S8* was applied by negative water column,

connecting a water-filled tube to the water outlet below the porous stone. Negative water columns were used instead of axis translation since variations of 1-2 kPa in the pressurized air supply line represented a relatively large inaccuracy for suctions <10 kPa in these load steps. After over-night consolidation before shear phase in load step 2 in tests *Silt S4* and *Silt S5*, the test chamber was moved horizontally until contact with the horizontal piston was re-established. The horizontal contact load was kept low until vertical net stress was re-applied, to avoid damaging the specimen.

**3.4 Load paths during shear tests**

The main motivation of the testing program was to investigate effect of suction on the shear strength. Consequently, shear tests were generally performed at constant vertical net stress while matric suction was increased stepwise. In each load step, either vertical net stress or matric suction was increased, never both, in order to separate the effect of each stress variable on the resulting shear strength. Load paths are illustrated graphically in Fig. 4. Table 2 gives an overview of testing conditions for all

shear tests. Four of the tests, *Silt S4*, *Silt S6*, *Silt S7* and the first five load steps of *Silt S8*, were performed at constant vertical net stress $\sigma_{v,net}$ of 18-24 kPa, which approximately equals in situ vertical net stress at 1-1.5 m depth. The effect of increasing vertical was investigated in the two ultimate load steps in test *Silt S8*, in order to determine the effective friction angle $\varphi'$. Test *Silt S5* was performed at $\sigma_{v,net}$ equal to 44 kPa, approximately twice the in situ vertical net stress. For load steps where matric suction was controlled with negative water column, $u_a=0$ and $u_w<0$. For load steps where matric suction was controlled by axis

translation, $u_a>0$, $u_w>0$, and $u_a>u_w$.

**4 Experimental results**

**4.1 Soil classification**

Grain size analysis (dry sieving) classifies the sample soil as sandy silt, very well sorted. The silt sample from 1.06-1.31 m depth contained 17 % sand and 83 % silt, with a gradation number $C_u$ of only 1.3. Table 1 summarizes volume/weight relations



for the shear box specimens. The porosity $n$ for all specimens varied within a narrow range, 43.6-45.1 % (corresponding to a void ratio $e$ of 0.77-0.82), corresponding to saturated gravimetric water content $w$ of 30+/-1 % for all specimens tested.

### 4.2 Saturated permeability

Saturated permeability on intact specimens of silt and sand was investigated in a triaxial apparatus by measuring upward flow through the specimen. Constant flow gradient was kept by a GDS pump at the lower end and atmospheric pressure at the top end of the specimen. For sandy silt, a value of approx. $1.0 \times 10^{-6}$ m/s was found (Fig. 5a) while a value of $1.0 \times 10^{-5}$ m/s was found for silty sand (Fig. 5b). Values considered relevant for permeability in situ were found in test steps 1-3 of each test, at small radial (cell) pressure. For higher radial stress (steps 4-6 in both tests) measured permeability decreases, assumedly corresponding to compaction of the specimens.

### 4.3 Retention curve

Prediction models presented in Section 2 use the retention curve or air entry value to predict shear strength as a function of suction. The retention (drying) curve for the sandy silt tested was investigated (Fig. 6), through mercury intrusion porosimetry, psychrometer and high capacity tensiometer measurements and negative water column, supplemented by predictions from granulometry (Heyerdahl and Pabst, 2017). The suction $S_{wr}$ corresponding to residual water content is equal to 10000 kPa for a gravimetric water content $w = 1.5$ %, applying the Fredlund and Xing (1994) method.

### 4.4 Results from unsaturated multi-stage shear tests

Representative tests results are shown for tests *Silt S7* (Fig. 7) and *Silt S8* (Fig. 8). Generally, shear strength increases with increased suction. The effect is highest for low suctions. In test *Silt S7* dilation is observed in all suction load steps, but not for the saturated first load step. Test *Silt S8* was performed as a multi-stage test along two stress state variables (Fig. 4). First, matric suction was increased in five steps, starting with saturated specimen. Thereafter, vertical net stress was increased in two additional load steps to determine the effective friction angle $\varphi'$ (section 5). Compression observed at the beginning of the first load step turned to dilation before peak shear stress. Dilation continued monotonically through subsequent load steps with increased matric suction. For increased vertical net stress in the two last load steps, vertical compression reoccurred, also during the shearing phase. In test *Silt S4* vertical compaction turned to dilation during shearing in the first load step, and dilation continues in the following two load steps. In test *Silt S5* vertical net stress was approx. twice the in situ stress (Fig. 4). Compaction was observed in the first two load steps, but turned to dilation in the third load step. In test *Silt S6* dilation was observed in all load steps.

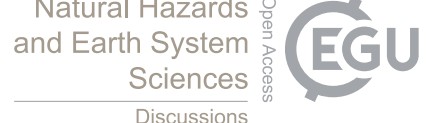


## 5 Discussion

As load increments in each load step were applied either for vertical net stress or matric suction, never for both stress state variables simultaneously (Fig. 6), effects of each of the stress state variables may be evaluated.

### 5.1 Effect of vertical net stress

Fig. 9 shows shear strength (peak shear stress) versus vertical net stress for all load steps. In load steps 6-7 in test *Silt S8* vertical net stress was increased. The resulting linear shear strength increase corresponds to an effective friction angle $\phi'$=36.9°. Higher vertical net stress in test *Silt S5* results in higher shear strength than tests at lower vertical net stress. A direct comparison between all tests is however not possible from Fig. 9, since applied suction varies. The suction effect appears from variation in peak shear strength at constant vertical net stress for each test. This variation illustrates possible effects of climatic factors, i.e. varying water content and suction, on the shear strength.

### 5.2 Effect of matric suction

To properly compare suction effects for tests performed at varying vertical net stress, measured peak shear stress values were translated to the zero vertical net stress-plane (Fredlund et al., 1987), assuming a constant effective friction angle of $\varphi'$=36.9° as interpreted from test *Silt S8*. Translation to the zero vertical net stress plane reduces spread in data (Fig. 10). Increased suction results in increased shear strength, but the relation is highly non-linear. The initial inclination of the shear strength curve is high, $\phi^b$ = 69.5° for test *Silt S7* (for suction 0-9.3 kPa), and $\phi^b$ = 58.3° for test *Silt S8* (for suction 0-7.8 kPa). For further suction increase, $\phi^b$ drops gradually in all tests. For suctions above 30-50 kPa, $\phi^b$ is practically equal to zero, and slightly negative for tests *Silt S6* and *Silt S7*. An effective cohesion (shear stress intercept for zero vertical net stress and zero suction) of $c'$=22 kPa is interpreted from tests *Silt S7* and *Silt S8*, starting with saturated specimens. High cohesion may be attributed to cementation effects due to near-surface hydro-chemical processes. Cementation expectedly has less influence at larger depth, and cohesion is also expected to decrease for increasing sand content. For a sand specimen collected in the same sample pit, effective cohesion was interpreted to 8 kPa (Heyerdahl, 2016).

### 5.3 Correction for dilation and water content

Dilation was present in most test steps involving increased suction (Fig. 7 and Fig. 8). The "saw tooth" model (Houlsby, 1991) defines the dilation angle $\psi_d$ as follows:

$$\frac{\tau}{\sigma_n} = \tan \varphi' = \tan(\varphi'_{cv} + \psi_d) \tag{9}$$

in which $\sigma_n$ is normal stress on the shear plane, $\phi'_{cv}$ is the critical state friction angle (friction angle at large deformation / constant volume), and $\psi_d$ is the dilation angle. Observed friction angle $\phi'$ hence is the sum of critical state friction and dilation. For constant vertical net stress, dilation equals the tangent of measured vertical displacement curve around failure ($\tau_f$).





Interpreted dilation angle $\psi_d$ shows considerable spread, with a maximum of 12°-14° for test *Silt S4*. Compaction was observed in test *Silt S5*, in which vertical net stress was higher than in the other tests. For increased vertical net stress in final steps of test *Silt S8*, dilation in preceding steps was turned to compaction (Fig. 5). Correction for dilation is not "necessary", as dilation is a real soil property present in the tests (but affected by test conditions). However, for comparison with shear strength

prediction models based on effective strength parameters, saturation rate and applied suction, and not taking dilation explicitly into account, a correction could be in place when comparing prediction models and laboratory data. One other argument supporting correction is that dilation may be a function of overconsolidation and net stress level. For the majority of data points with positive dilation, vertical net stress however was comparatively low (< 25 kPa). Corrections for dilation is therefore moderate and does not change main conclusions (Fig. 10).

For multi-stage shear tests there is an inherent uncertainty about actual water content during each load step, since water content may not be measured during the test. Circulation of pressurized air, generally not fully saturated with vapour, is a potential source of error, as well as possible air leakages from the test chamber, e.g. around the vertical load piston. A review of water contents in the specimens for each load step was done based on water contents after tests. Based on measured water flow, water contents in each test step were back-calculated, taking air diffusion into account (Heyerdahl, 2016). Water contents were

compared with the retention (drying) curve (Fig. 6). Generally, water contents after tests were higher than given by the retention curve, not lower, as expected from potential error sources mentioned above. One reason may be too short consolidation time before shearing (some load steps in tests Silt S4-S6); however, high water contents compared to the retention curve were observed also in time steps with over-night consolidation and more. Relating back-calculated water contents of each load step to corresponding suction values from the retention curve, data points move towards lower suction values, and the plot of

measured $\tau_f$ values collapses further (Fig. 11).

**5.4 Comparison with prediction models**

Shear test results were compared with prediction models presented above (Fig. 12). Vanapalli 1st and 2nd models predict an initial increase in shear strength, followed by a reduction in shear strength. The strength reduction starts around air entry value of 8 kPa for the Vanapalli 1st and approx. 20 kPa for the Vanapalli 2nd model. Reduction is gradual up to ~40 kPa for both

models. For higher suction predicted strength increase is negligible or very small, i.e. increased suction has no further effect. For Vanapalli 1st the shear strength stabilizes at a constant value approx. equal to the effective cohesion. Both models predict far lower shear strength than measured in the tests, in agreement with Kim and Borden (2011). The Khalili-Khabbaz model gives about similar match with test data for matric suction up to 10-20 kPa, but better match than the Vanapalli methods for higher suctions. Generally, measured peak shear stress however is considerably higher than predicted values for all these

prediction models, which at best seems to indicate a lower bound for the unsaturated shear strength. Changing the exponent for $\chi$ in the Khalili-Khabbaz formulation (Eq. 6) from the suggested value $\chi = -0.55$ to $\chi = -0.25$, and then to $\chi = -0.15$ (Fig. 13), resulted in improved shear strength prediction in the suction range applied in these tests. However, predicted shear strength increases rapidly for increased suction. Shear tests at higher suction would be necessary to verify the tested exponent values.





### 5.6 High value of $\phi^b$

Values of $\phi^b \gg \phi'$ does not agree with the general assumption of $\phi^b = \phi'$ below air entry value, but such values are still found in literature: Unsaturated shear strength of two clayey colluvial soils from Brazil increased linearly with $\phi^b > \phi'$ for suctions up to 150 kPa (Feuerharmel et al., 2006). The shear strength increased along the horizontal part of a retention curve (i.e. constant

water content), verifying that increased suction was responsible for shear strength increase. Values of $\phi^b > \phi'$ for Brazilian residual soils were also found by Röhm (1995). The results may indicate that dilation was present in the reported tests. Dilation during suction shearing has been reported also by other authors, e.g. in unsaturated shear tests on loosely compacted, completely decomposed granite for low vertical net stress (Chen et al., 2004). Stress-path dependent dilation was also observed by Ng and Chiu (2001) for unsaturated strength of loosely compacted volcanic soil ("slightly sandy silt"). Hussain (2010)

observed dilation for low vertical net stress in shear tests on unsaturated specimens of compacted, completely decomposed, granite, also at saturated conditions. Similar effects on shear strength of unsaturated sand specimens were reported by Shwan and Smith (2015), assuming that measured peak strength values above expected values might be related to dilation; however, vertical displacement measurements were not reported. Röhm (1995), Feuerharmel et al. (2006) and Hussain (2010) reported increased effective friction angle for increased suction. An increase in $\phi'$ may be observed as an increased $\phi^b$. Increased $\phi'$ has

been attributed to structural changes causing increased grain interlocking, e.g. Campos (1997). Ng and Chiu (2001) and Laloui (2010), on the other hand, conclude that internal friction appears independent of suction. Reported effects of grain interlocking may also increase the tendency to dilation for suction shearing. Micro-mechanical analyses show that water menisci produce intergranular stress at the grain-to-grain contacts (Tarantino, 2010). This stress may exceed saturated intergranular stress, is almost independent of matric suction and acts like an on/off effect: When the soil gets saturated, menisci and intergranular

suction stress at grain-to-grain contacts will disappear. Menisci will not be present on a large scale until some water has been evacuated from the specimen, but when menisci are present, contractive forces will appear and be almost constant. Increased shear strength should therefore be observed when menisci are present, i.e. around the air entry value. Micro-mechanical effects are not included directly in commonly used prediction models, as in Vanapalli et al. (1996b).

Based on the measured results and the discussion above, it is concluded that dilation is a property of the tested soil, although

the effect of dilation in the actual tests is comparatively small (Fig. 10). Dilation caused by suction will have increased relative importance for reduced vertical net stress, i.e. it may be important in the triggering process of shallow landslides.

### 5.7 Suction overconsolidation

Changes in net stress and pore-water pressure both have effect on unsaturated shear strength, but are different loading conditions: Pore-water pressure (positive or negative) is isotropic, while in situ net stress is anisotropic; normally with larger

vertical than horizontal net stress. Decreased saturation / increased suction will produce inter-granular forces concentrated at menisci as grain-to-grain contacts. Pore-water pressure in saturated soils acts around the entire soil particle. As discussed above, dilation may result from contractive forces in menisci around grain-to-grain contacts. Differences could therefore be



expected for the effect of numerically equal changes in suction and e.g. vertical net stress, resulting in different response in shear strength (and volumetric behaviour). Effect of previously experienced suction is included in e.g. constitutive models as the Barcelona Basic model ("BBM") where reference suction $s_0$ or $(u_a - u_w)_0$ is defined as the maximum past suction experienced by the soil (Alonso et al., 1990). The SI (suction increase) yield locus encloses an elastic region, while stress

trajectories crossing (i.e., expanding) the yield locus will result in irreversible strains. Suction results in a stiffer volumetric response within than outside the pre-consolidated area. Suction overconsolidation could therefore be expected to manifest itself as increased shear strength. Inclusion of suction overconsolidation in prediction models for unsaturated shear strength seems rational, but requires more experimental evidence than available from the tests reported herein.

### 6 Numerical analyses of slope during extreme long-term rainfall, autumn of year 2000

**6.1 Model slope and analyses**

Numerical analyses were performed for the slope that failed in year 2000 (Fig. 1b), taking unsaturated soil properties into consideration for calculation of infiltration and slope stability. The geotechnical software package GeoStudio (Geo-Slope_International, 2015) was used (sub-packages Seep/w for unsaturated flow analysis and Slope/w for stability analysis). Analyses were done for two alternative model slopes. A 6-7 m thick top layer of alternatively sand or silt is modelled on top

of a thick clay layer (Fig. 16). Effective strength parameters are $\phi'$=41.0° and $c'$=8 kPa for sand (Heyerdahl, 2016), and $\phi'$=36.9° (Fig. 9) and $c'$=22 kPa (Fig. 10) for silt. A high effective cohesion was applied in clay to keep failure surfaces within the top layer, as observed for the actual slope failure in year 2000. Suction contribution to unsaturated shear strength is calculated from the product of water content and suction from seepage analyses. Test data from unsaturated tests on silt specimens presented above, as well as data from unsaturated tests on sand from the same site (Heyerdahl, 2016), confirm that this

assumption may be considered the minimum unsaturated strength to be expected for these soils.

The wetting branch of the retention curve (Fig. 14) was predicted from measured retention drying curves (Heyerdahl and Pabst, 2017), applying the "Universal Mualem" formulation (Mualem, 1977). Permeability functions for infiltration and flow analysis (Fig. 15) were constructed from wetting curves and scaled with saturated permabily (Fig. 5). The flow calculation includes the clay layer, with assumed saturated permeability of $1 \times 10^{-9}$ m/s, which gives very low flux through the clay.

Potential infiltration (boundary condition at the soil surface) was based on regional rainfall prior to slope failure. The autumn of year 2000 gave unusually large long-term rainfall in Eastern Norway, comparing to return periods well above 100 years for monthly rainfall (Jaedicke and Kleven, 2007). In the first step of the seepage analysis, annual rainfall of 800 mm was distributed evenly, to approximate initial conditions for analysis of subsequent heavy rainfall. Infiltration is influenced by distribution of rainfall in time, and may be reduced by evaporation, runoff of surface water or biologic activity. Modelled annual groundwater

however serves as an initial situation. Based on historic rainfall records from year 2000, the second step of the seepage analysis was performed, applying continuous rainfall of 240 mm distributed evenly over 30 days (8 mm/day). Excess water ponding at the slope surface was removed between time steps in the seepage analysis (surface runoff).





Slope stability was calculated for average annual rainfall and for extreme rainfall in year 2000, including variation during the 30 day rainfall period. Some variations in effective soil strength (cohesion) were performed.

### 6.2 Results of numerical analyses

#### 6.2.1 Sand slope

For a model slope with top layer of sand, annual rainfall 800 mm results in a groundwater line approx. 1 m above the clay layer (Fig. 17a). Above groundwater, pore-water pressures are negative (i.e. suction). During the year 2000 rainfall groundwater rises to approx. 3 m above the clay layer (at the middle of the plateau). Pore-water pressures during the year 2000 rainfall are presented for a vertical profile at the slope crest (Fig. 17b). The first 5 days the pore-water pressure distribution resembles a vertically propagating infiltration front, with practically no change below 3 m. After 30 days, pore-water pressure

in profile P1 has increased by approx. 10 kPa throughout the sand layer, but there is still suction down to 5 m depth (Fig. 18). The slope safety factor for the sand slope for annual rainfall is $\gamma_m=2.59$ (Fig. 17a). The critical failure surface is approximately 5 m high and 2 m deep, i.e. quite similar to the failure in year 2000 (Fig. 1b). After the year 2000 rainfall, the safety factor drops to $\gamma_m=2.25$, which in general represents good stability (Fig. 17b).

#### 6.2.2 Silt slope

For a model slope with top layer of silt, the groundwater for annual rainfall lies 3-4 m above the clay layer (Fig. 19a). Above groundwater, pore-water pressures are negative. The year 2000 rainfall saturates the slope to the surface (Fig. 19b). The pore-water pressure increases almost equally much at the slope surface and at the bottom of the silt layer (Fig. 20). The pore-water pressure increases by 15-20 kPa during the rainfall, and all soil suction dissipates.

For annual rainfall, the safety factor $\gamma_m=2.74$ (Fig. 19a), while for the year 2000 rainfall the resulting safety factor of $\gamma_m=1.70$

(Fig. 19b).

#### 6.2.3 Reduced cohesion $c'$ in sand and silt

For both silt and sand slope, cohesion is responsible for a large portion of the safety margin. Measured cohesion values $c'=8$ kPa in sand and $c'=22$ kPa in silt are comparatively high for granular/intermediate soils with low fines content. Cohesion may possibly be due to cementation of grains in shallow layers. For failure surfaces 4-5 m below the soil surface, lower cohesion

might be expected.

Reducing cohesion to $c'=1$ kPa in sand results in slope safety factor $\gamma_m=1.90$ for annual rainfall (Fig. 21a). Keeping the failure surface fixed, the safety factor drops continuously during the year 2000 rainfall to $\gamma_m=1.61$ after 30 days (Fig. 21b). Searching for the most critical failure surface, a shallow failure mode under groundwater was found critical, and reached $\gamma_m=1.25$ during the year 2000 rainfall (Fig. 21c). The slope is theoretically safe after the year 2000 rainfall, but the safety factor drops for both

failure modes (Fig 22).





By reducing cohesion to 5 kPa in the silt layer, the safety factor drops to $\gamma_m$=1.43 for annual rainfall (Fig. 23a), and to $\gamma_m$=1.02 after the year 2000 rainfall (Fig. 23b); in other words, the slope is practically at failure. The geometry of the critical failure is approximately as the actual slope failure in year 2000 (Fig. 1b). After 19 days of rainfall during the autumn of year 2000, the slope is fully saturated (Fig. 22). After this point, the slope stability remains constant $\gamma_m$=1.02).

**7 Conclusions**

Unsaturated shear tests on intact specimens of Quaternary sandy silt from Eastern Norway showed a substantial effect on shear strength for suction up to 20-25 kPa. Prediction models based on soil retention properties, as the Vanapalli $1^{st}$ / $2^{nd}$ and Khalili-Khabbaz prediction models, all underestimated the actual unsaturated shear strength, with the Khalili-Khabbaz model giving slightly better prediction. Observed shear strength increase $\varphi^b$ from applied suction exceeded the effective friction angle $\varphi'$ for

suctions up to 20-25 kPa. It is normally assumed that $\varphi^b=\varphi'$ (exceptions are however known from literature). Dilation observed during suction shearing only partly explains the observed high unsaturated shear strength.

Numerical seepage and unsaturated flow analyses with subsequent stability calculations were performed to study effects on slope stability of extreme long-term rainfall, as recorded the autumn of year 2000 in Eastern Norway. The results indicate that extreme long-term rainfall may have caused considerable rise in groundwater, and dissipation of soil suction compared to

"normal" ground-water conditions. The results indicate the cause of the landslide in year 2000 at Negarden Sander. However, to reach critical slope stability, cohesion values had to be reduced considerably. For the silt slope, cohesion of 5 kPa resulted in critical stability for fully saturated slope after the year 2000 rainfall. The sand slope was still theoreticallty stable for cohesion of 1 kPa, which means that longer rainfall (with same intensity) would be necessary to destabilize the slope.

During the last decade, rainfall events with volumes and intensities higher than measured historically have occurred several

20   times. These events have resulted in many landslides in Eastern Norway (Heyerdahl and Høydal, 2017). The work presented herein indicates that evaluation of slope stability based on unsaturated soil properties will be increasingly important when it comes to understanding rainfall triggering of landslides. Future work should include a wider spectre of soil types and geologic/topographic settings, as well as various types of preceding rainfall, rainfall intensities and durations.

**8 Acknowledgements**

The author is indebted to prof. E. Romero for allowing me access to the unsaturated soil testing facilities in the geotechnical laboratory, Catalonian Technical University in Barcelona (UPC). All other staff, co-students and research fellows during my stay at UPC are also thanked. A special thank goes to PhD Alessio Ferrari for introducing me to the unsaturated shear testing device and invaluable advice and encouragement through the laboratory program. A special thank also goes to Cand. Agric. Tore E. Sveistrup for teaching and helping with soil profiling and sampling. My supervisors, prof. Anders Elverhøi (Univ. of

Oslo), PhD Farrokh Nadim (technical director, NGI), prof. em. Kaare Høeg (Univ. of Oslo/NGI) and prof. Nils-Otto Kitterød





(NMBU, Ås), are all thanked for comments to the manuscript and support. Also PhD Finn Løvholt at NGI and Anders Solheim (director Natural Hazards, NGI/Adj. prof. Univ. of Oslo), are thanked for comments to the manuscript. Financial support received from the Norwegian Geotechnical Institute Research Fund is highly appreciated.

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





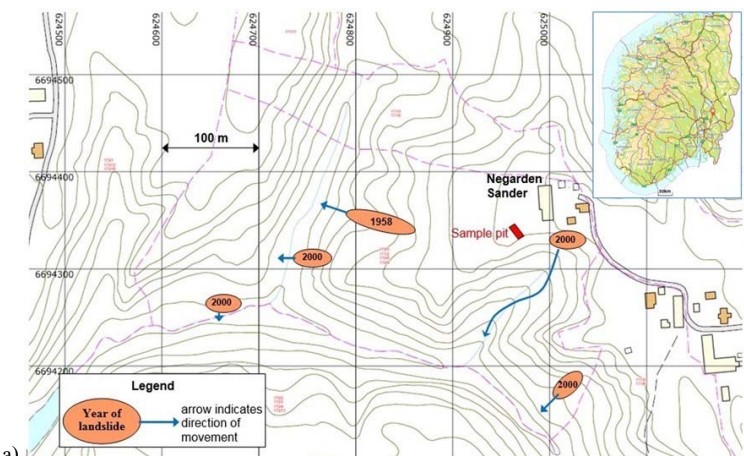

a)

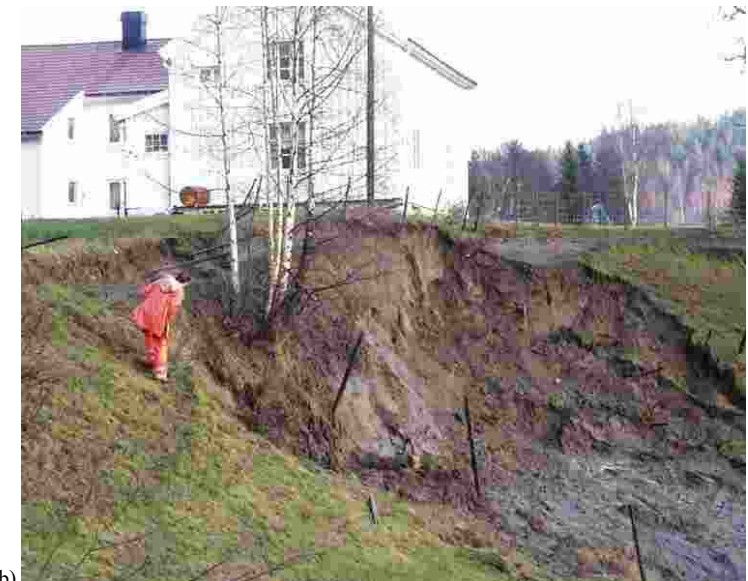

b)





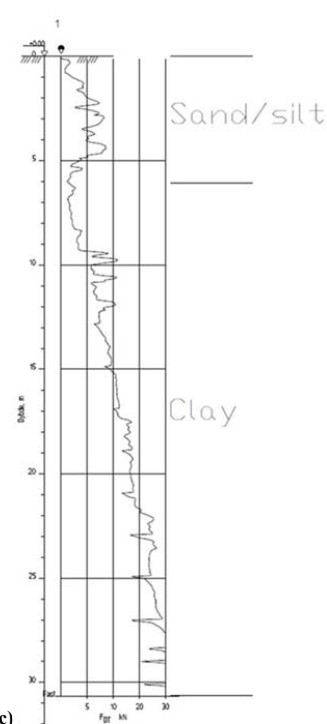

c)

**Fig. 1 a) Study area at Negarden Sander, municipality of Eidsvoll, Upper Romerike region in Eastern Norway (map data: http://kart.dgi.no/GISLINEWebInnsyn_dgi/) b) Landslide in silt/sand at Negarden Sander 22 November 2000 after long-term rainfall, south of farmhouse at Negarden Sander (Fig. 1a). c) Rotary pressure sounding at top of slope (close to sample pit)**





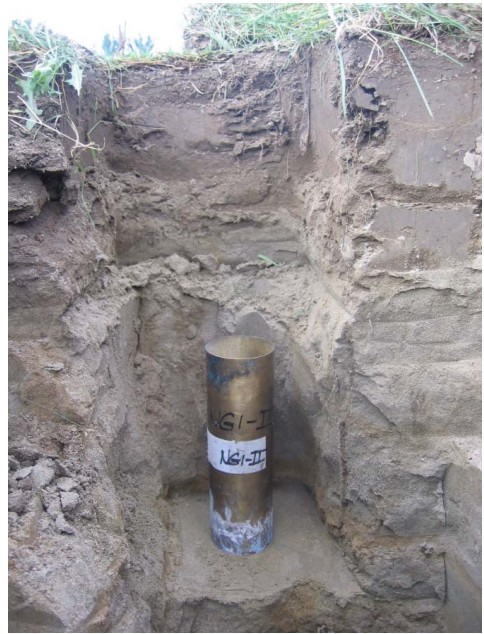

a)

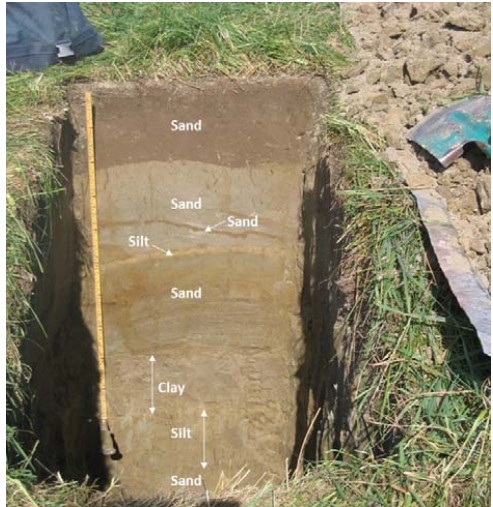

b)

**Fig. 2 Sample pit at Negarden Sander, Eidsvoll, Norway a) Sample cylinder ready to penetrate ïnto silt layer b) Main soil layers**





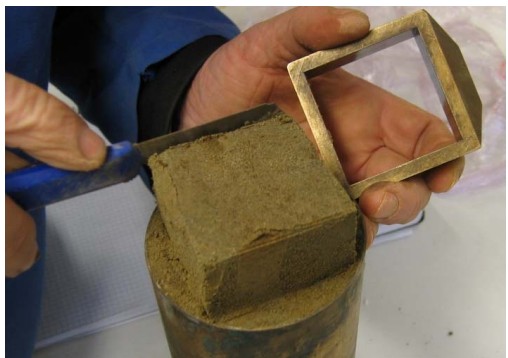

a)

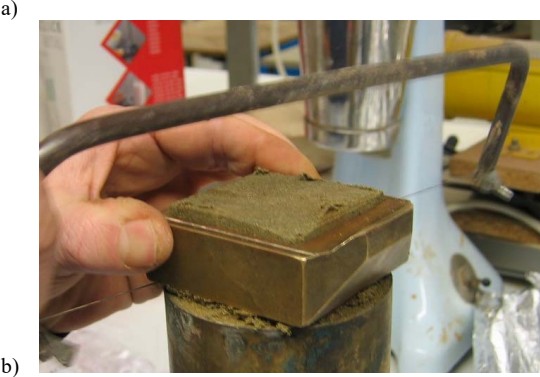

b)

**Fig. 3 Preparation of specimen for shear box test from cylinder sample a) Extruded part of sample cut into 50x50 mm square b) Top**
5   **half of shear box slid onto sample; specimen released by cutting underneath shear box with thread saw**





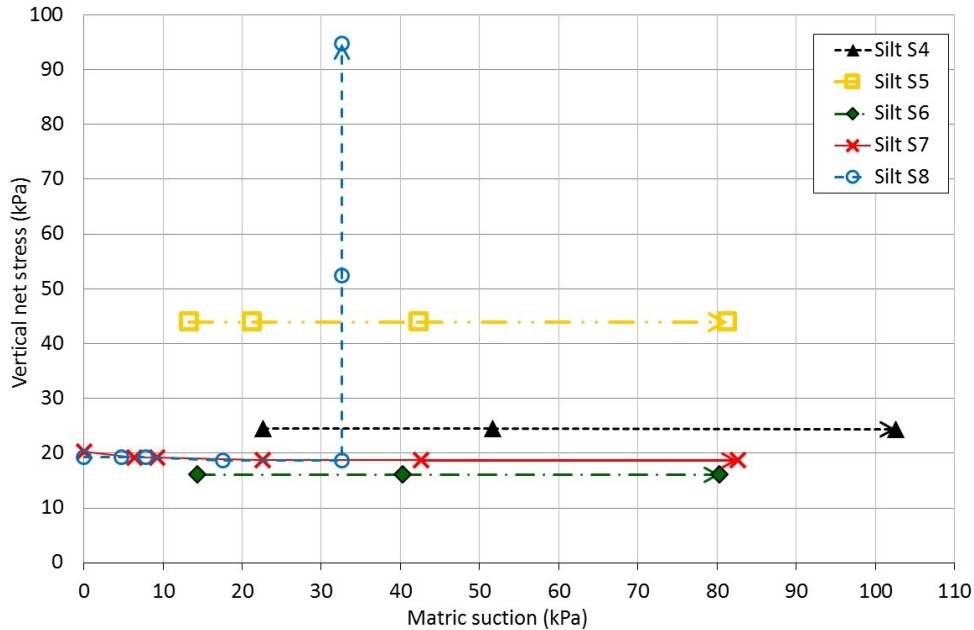

**Fig. 4 Load paths in direct shear tests**



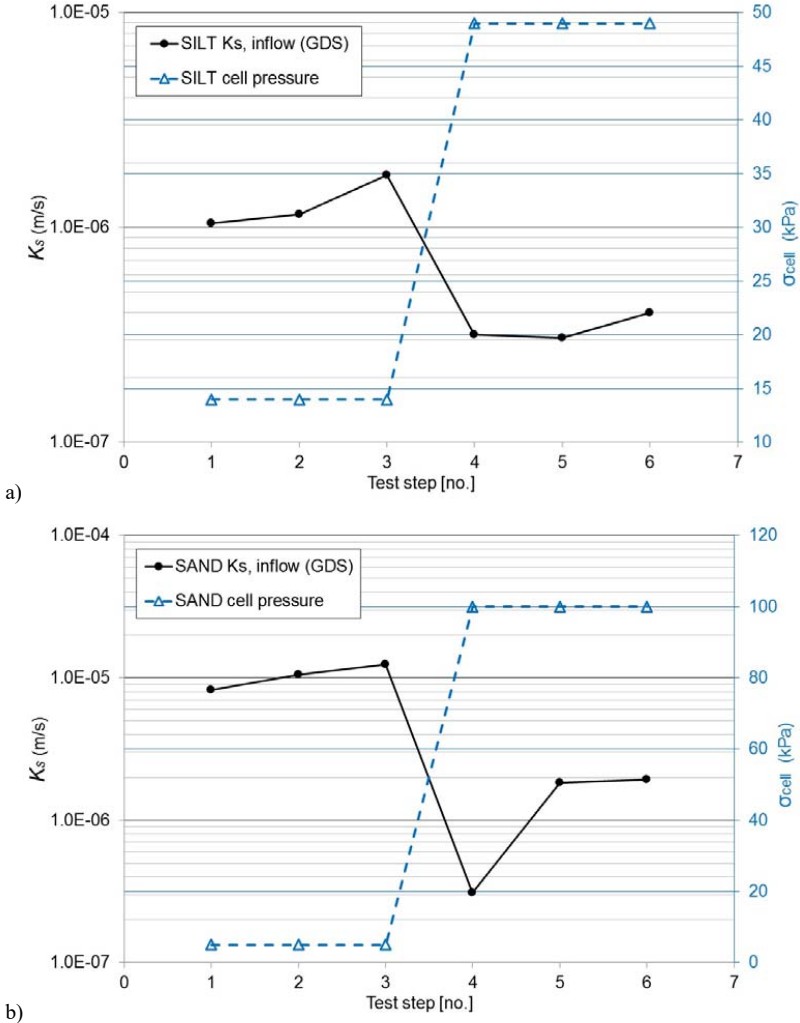

**Fig. 5 Permeability measurement on intact specimens in triaxial cell a) Silt b) Sand**



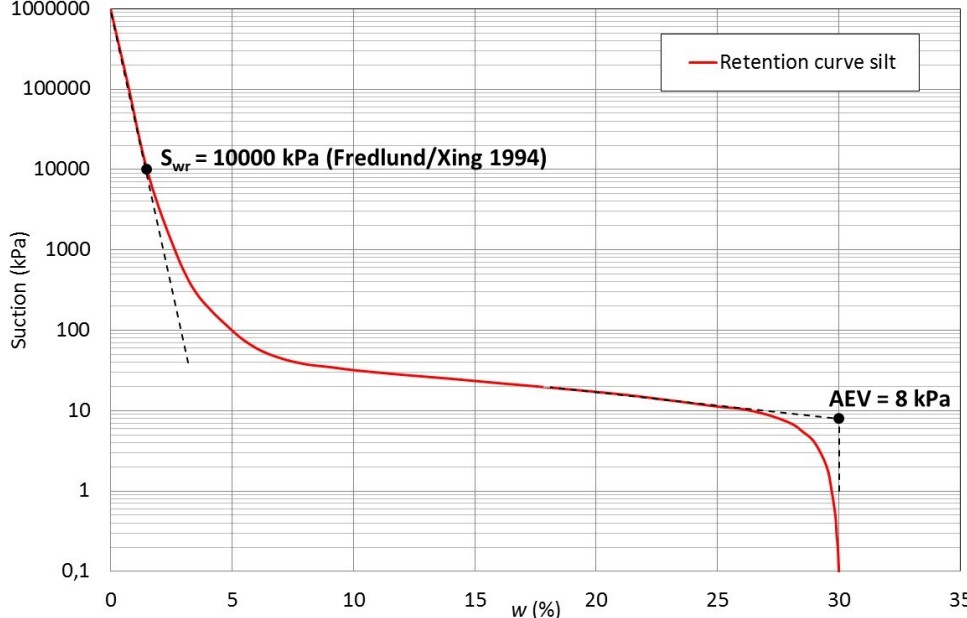

**Fig. 6 Retention curve (main drying branch) measured on specimens from the silt layer**




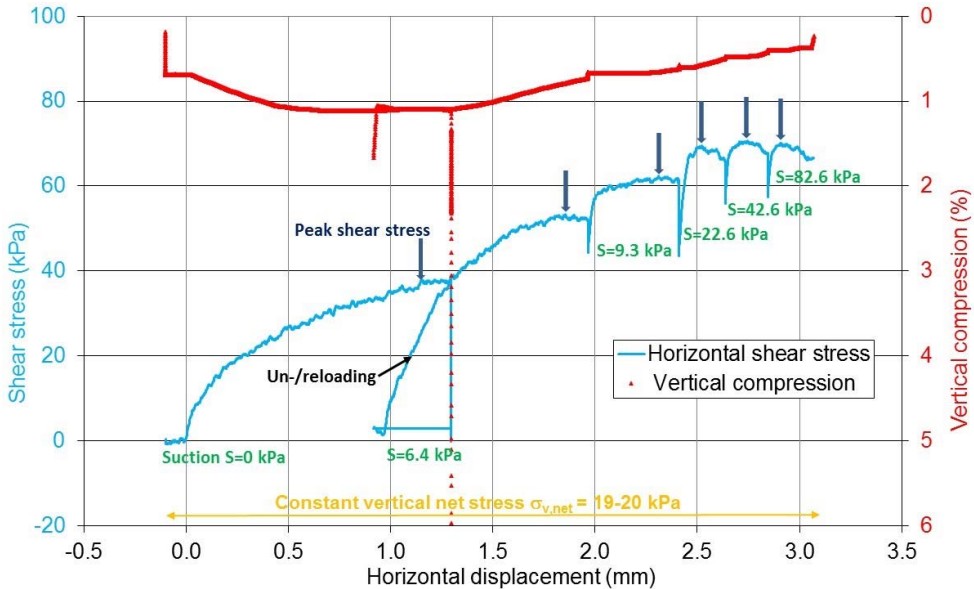

**Fig. 7 Direct shear test *Silt S7*.**



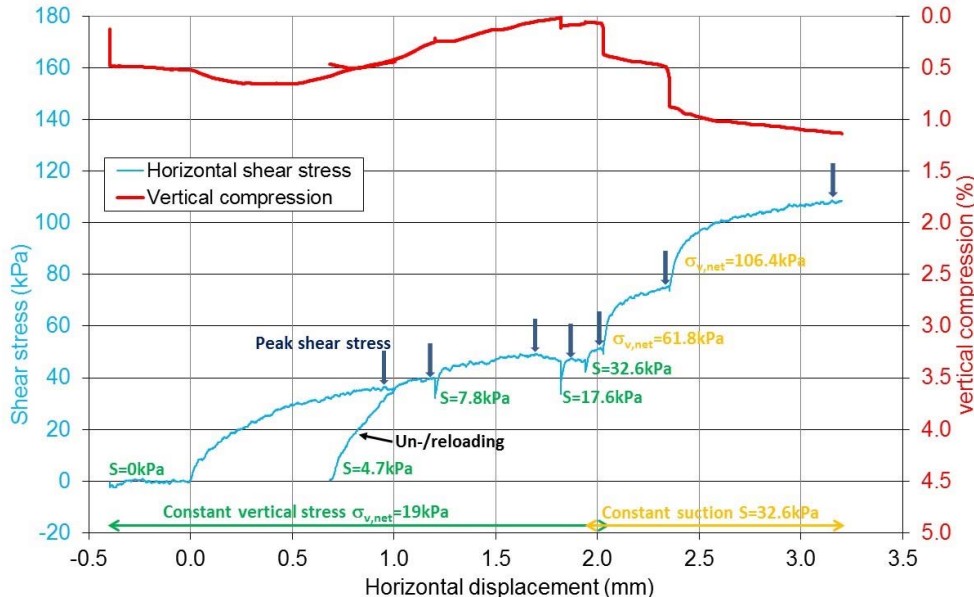

**Fig. 8 Direct shear test *Silt S8*.**





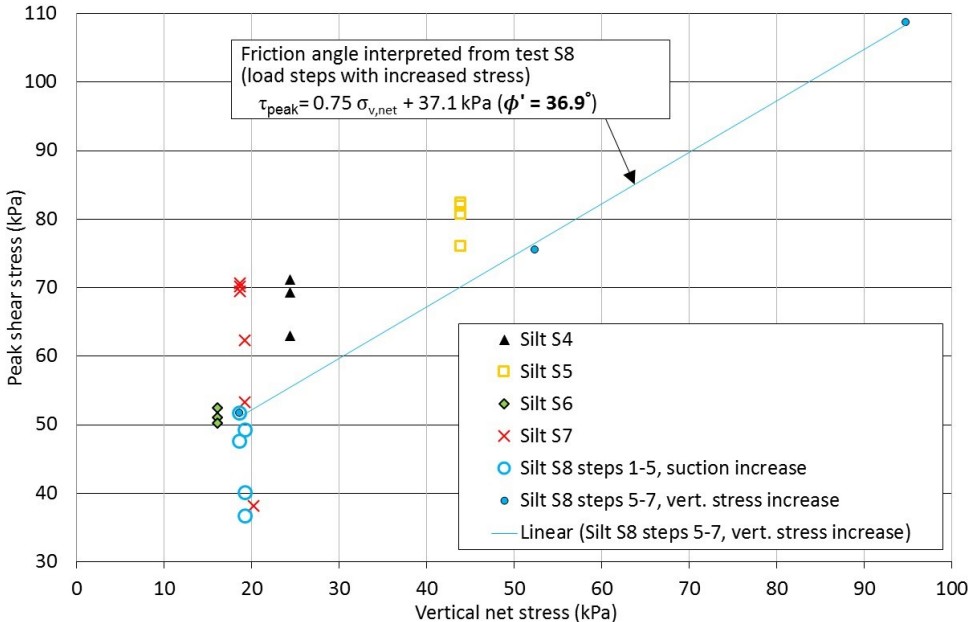

Fig. 9 Shear tests Silt S4-S8. Peak shear stress versus vertical net stress





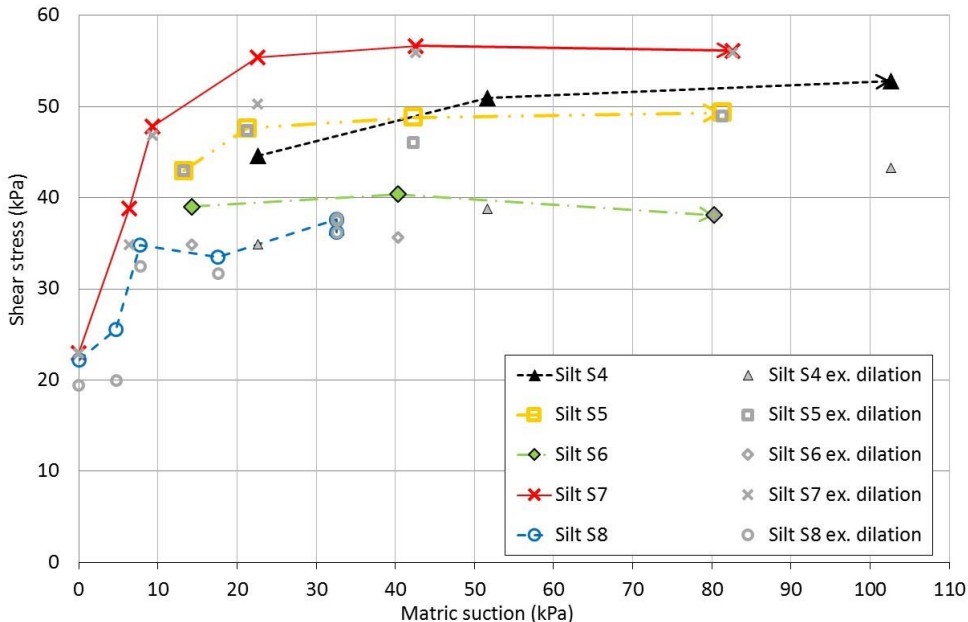

**Fig. 10 Shear tests Silt S4-S8. Measured peak shear stress translated to zero vertical net stress. Measured values corrected for dilation also shown.**





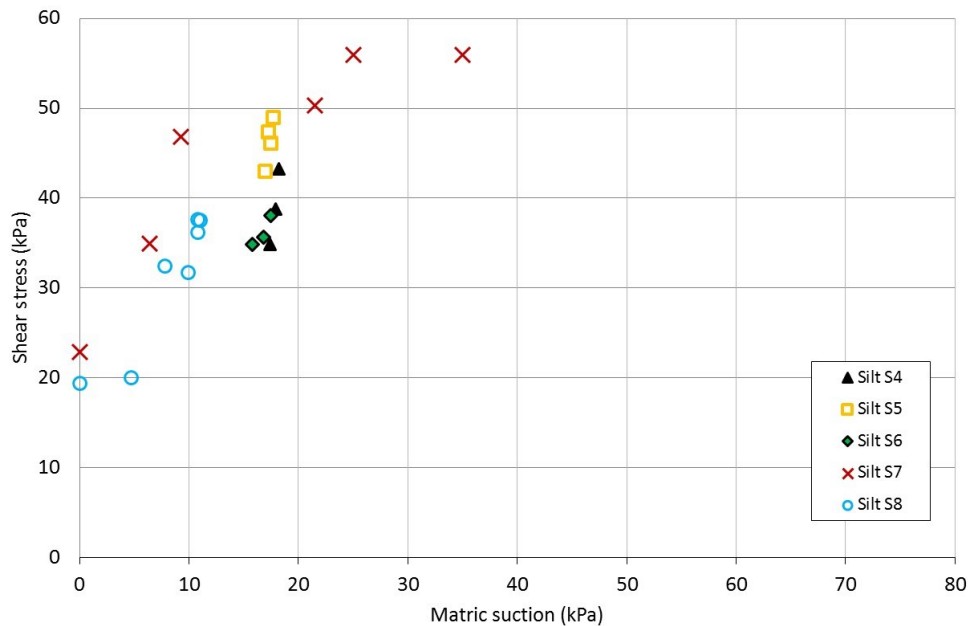

**Fig. 11 Shear tests Silt S4-S8. Measured peak shear stress translated to zero vertical net stress. Suction values corrected based on retention curve (Fig. 2) and measured water content at end of testing. Corrected for dilation.**





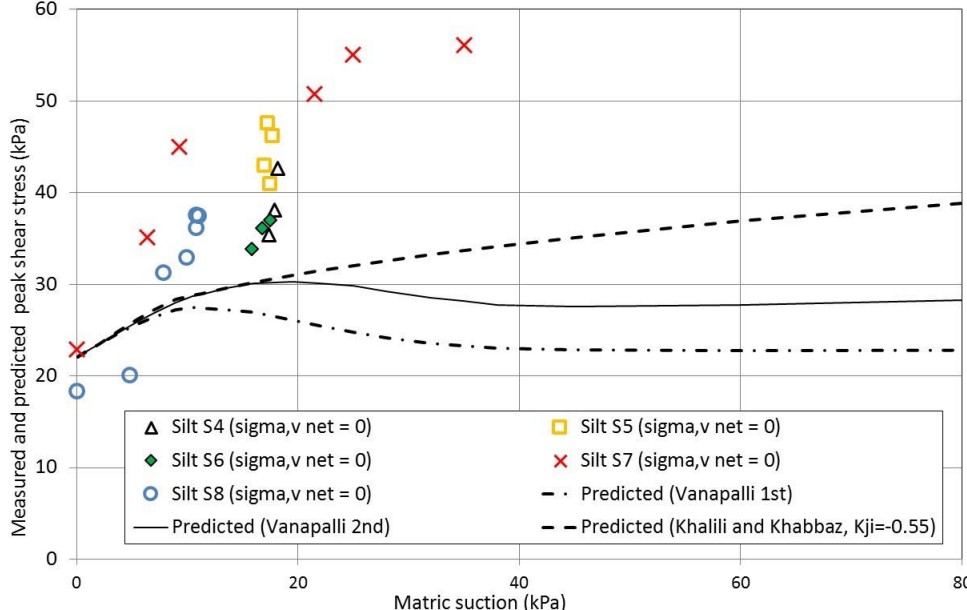

**Fig. 12 Comparison of measured peak shear stress and predictions with Vanapalli's 1st/2nd and Khalili-Khabbaz models for zero vertical net stress.**

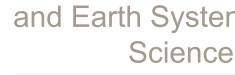
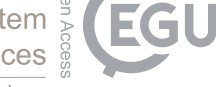


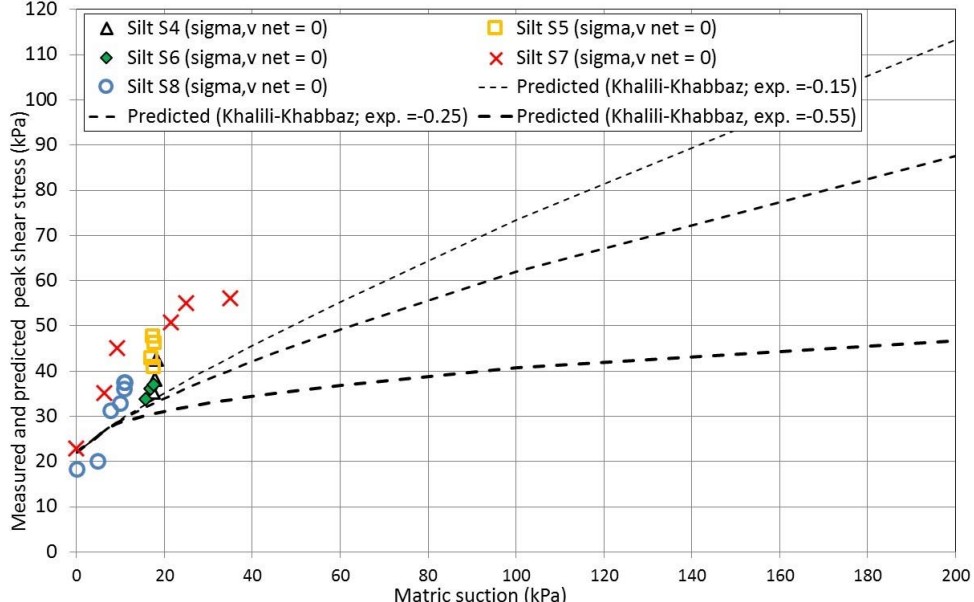

Fig. 13 Effect of varying the exponent in the Khalili-Khabbaz unsaturated shear strength prediction model (Eq. 6)





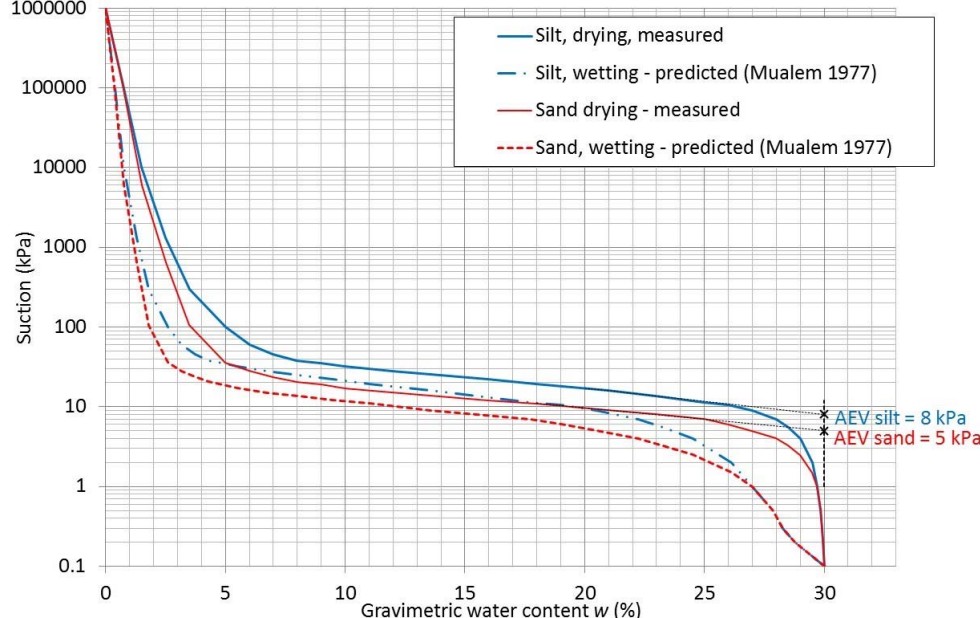

**Fig. 14** Retention curves for silty sand and sandy silt. Wetting branches predicted by the Universal Mualem method (Mualem, 1977) from drying curves measured in laboratory (Heyerdahl and Pabst, 2017).





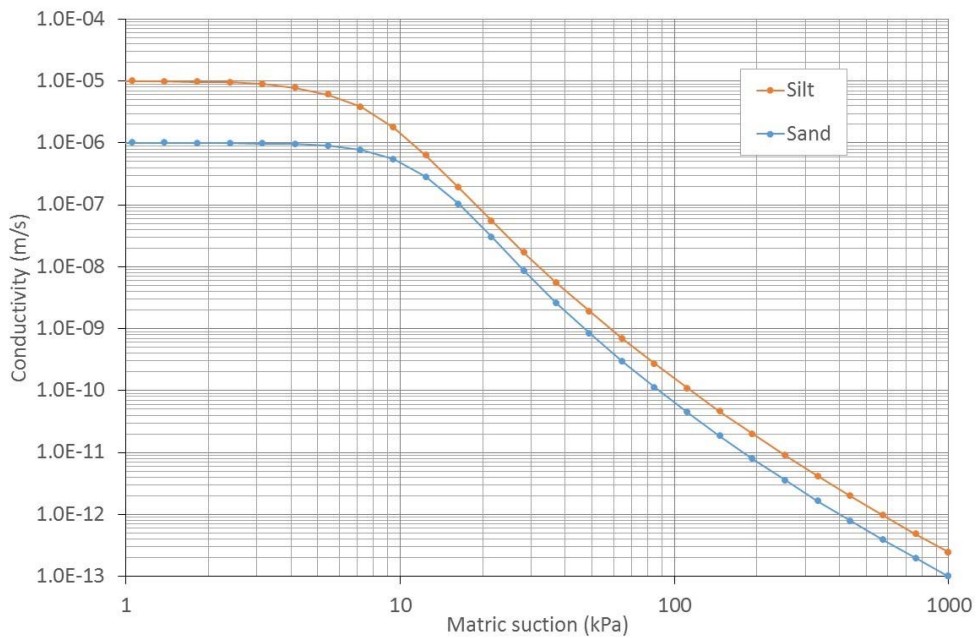

**Fig. 15 Calculated unsaturated hydraulic conductivity functions for wetting branch of retention curves.**

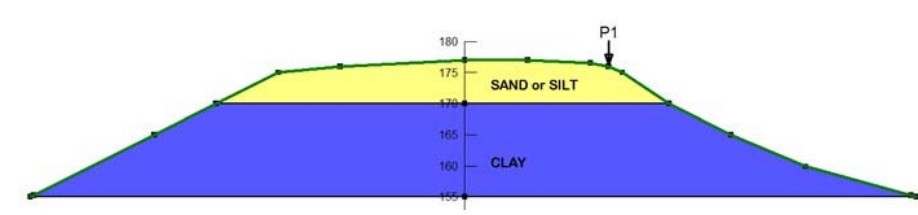

**Fig. 16 Model slope. Top layer of either silt or sand, on top of a thick clay layer**





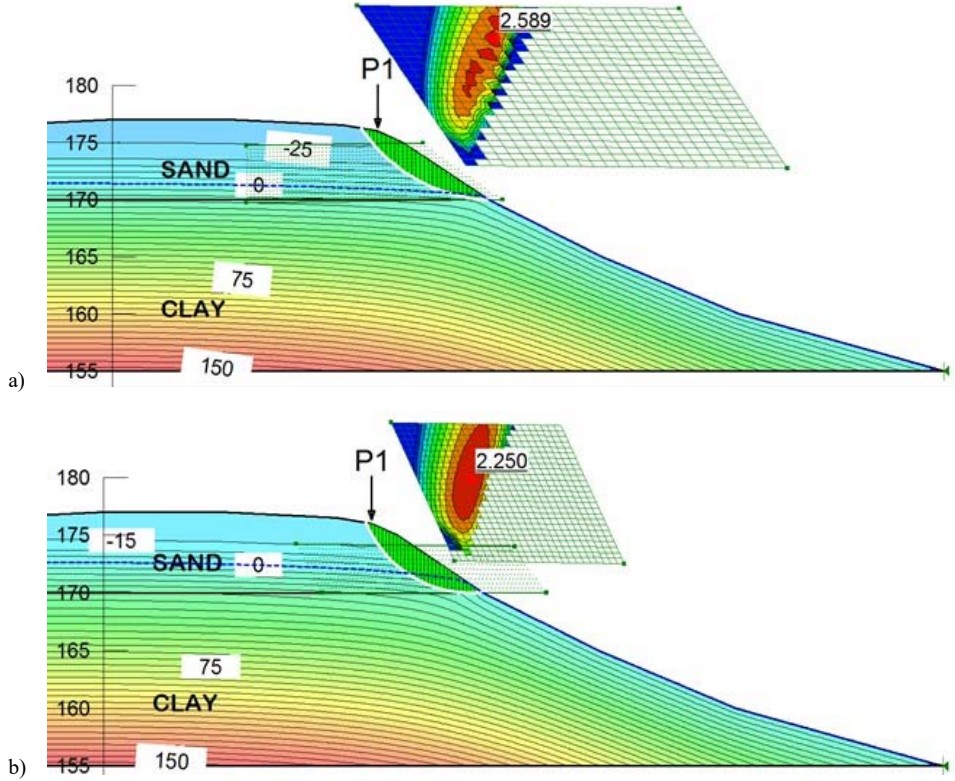

**Fig. 17 a) Pore-water pressure and slope stability for model slope of sand above clay. Normal annual rainfall (800 mm) b) Rainfall autumn of year 2000 (240 mm in 30 days)**





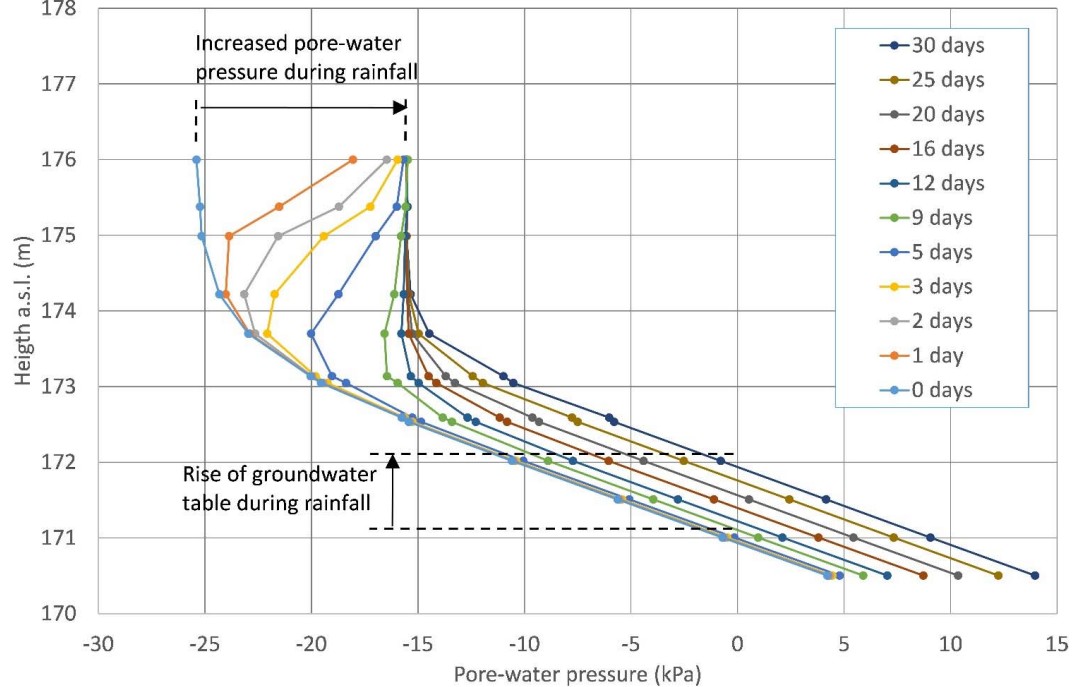

Fig. 18 Development of pore-water pressure in model slope of sand above clay (profile P1 in Fig. 19) during 30 days of extreme long-term rainfall (8 mm/day). Initial pore-water pressure distribution (t=0 sec) from average annual rainfall (steady-state)

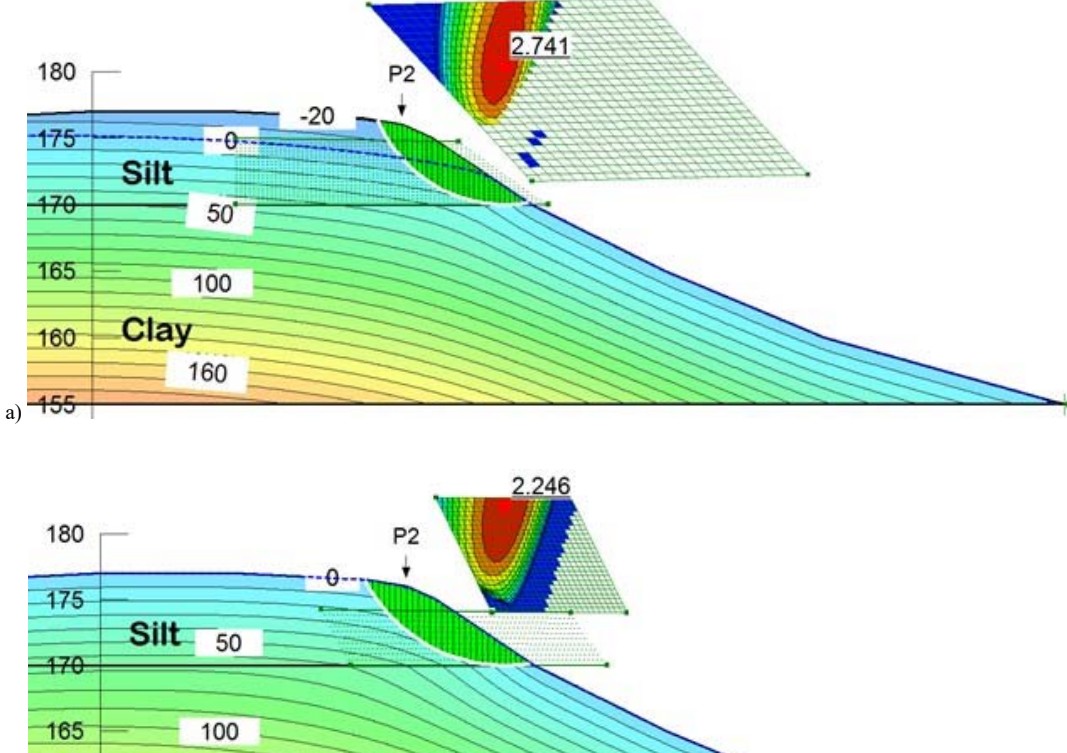

**Fig. 19 a) Pore-water pressure and slope stability for model slope of silt above clay. a) Normal annual rainfall (800 mm) b) Rainfall autumn of year 2000 (240 mm in 30 days)**



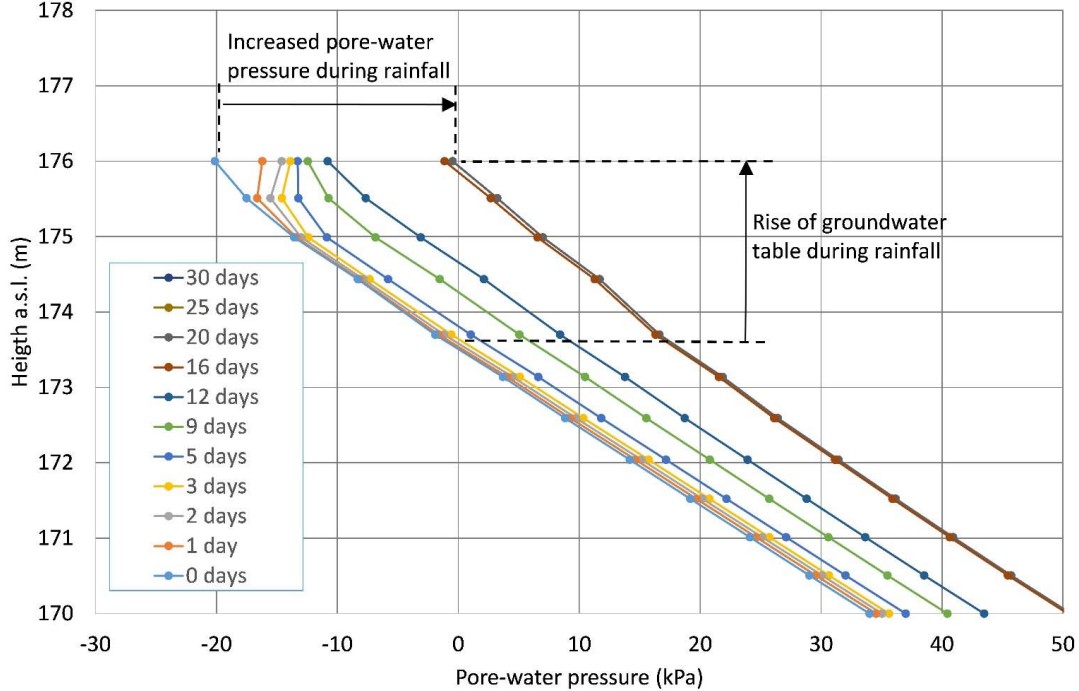

**Fig. 20 Development of pore-water pressure in model slope of silt above clay (profile P2 in Fig. 19) during 30 days of extreme long-term rainfall (8 mm/day). Initial pore-water pressure distribution (t=0 sec) from average annual rainfall (steady-state).**



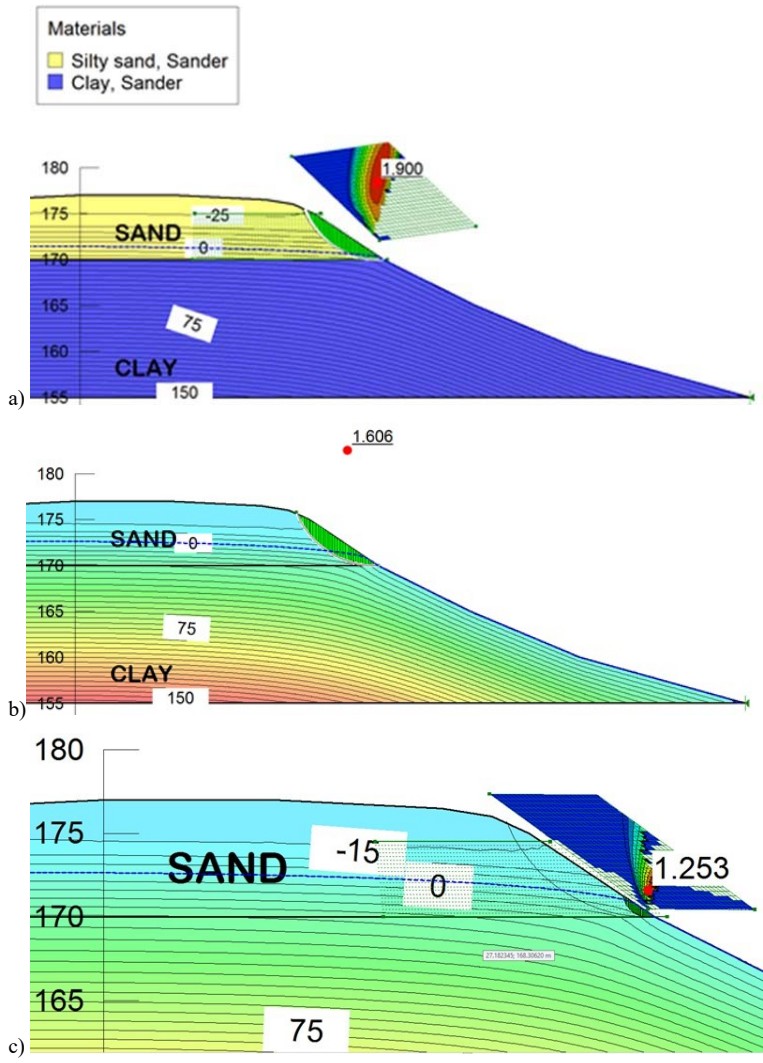

5  **Fig. 21 Model slope of sand above clay. Cohesion in sand $c'$=1 kPa a) Critical failure surface for annual rainfall b) Stability for year 2000 rainfall, critical failure surface from annual rainfall c) Critical failure surface year 2000 rainfall (below groundwater line)**





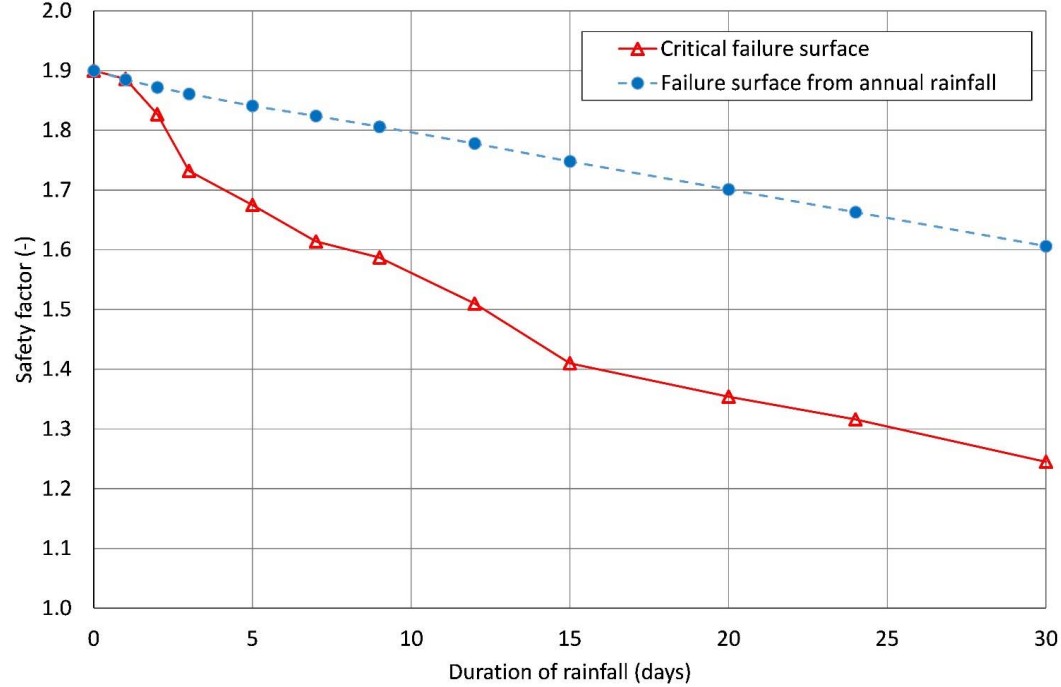

**Fig. 22 Model slope of sand above clay. Cohesion in sand c'=1 kPa. Development of slope safety factor during extreme rainfall autumn year 2000 (240 mm in 30 days)**





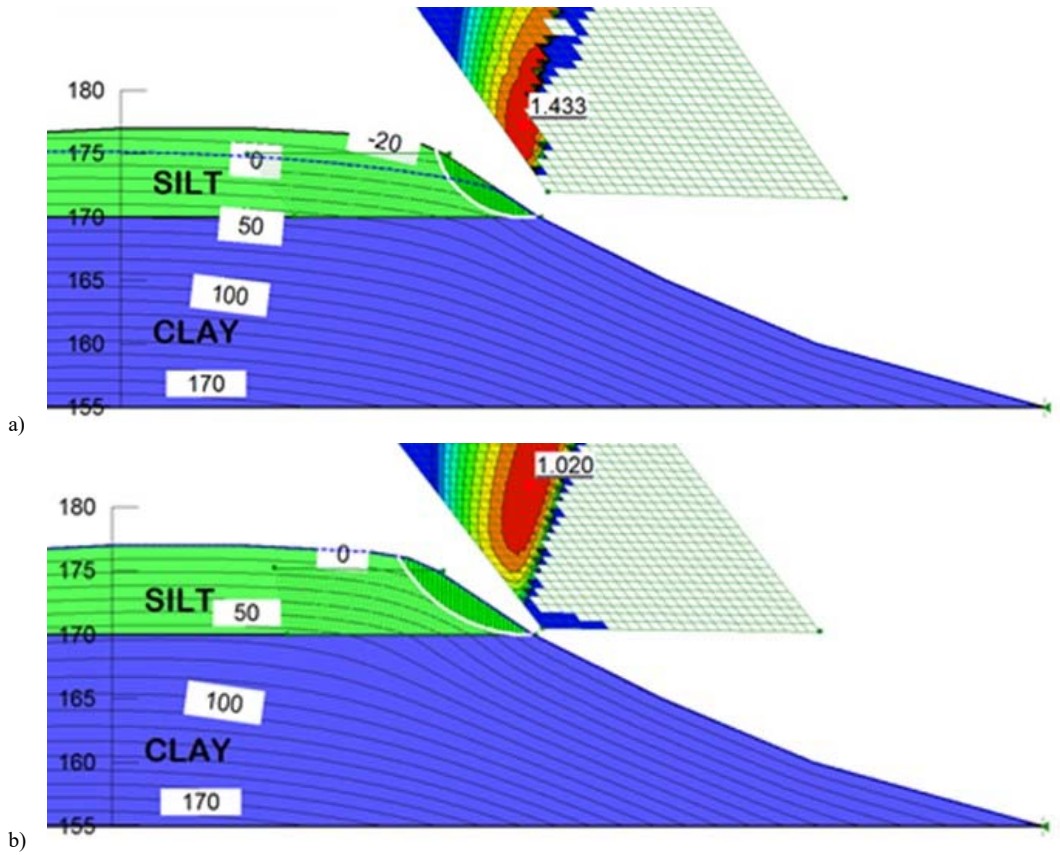

**Fig. 23 Model slope of silt above clay. Cohesion in silt c′=5 kPa. a) Critical failure surface for annual rainfall b) Critical failure surface for rainfall autumn year 2000 (240 mm in 30 days)**




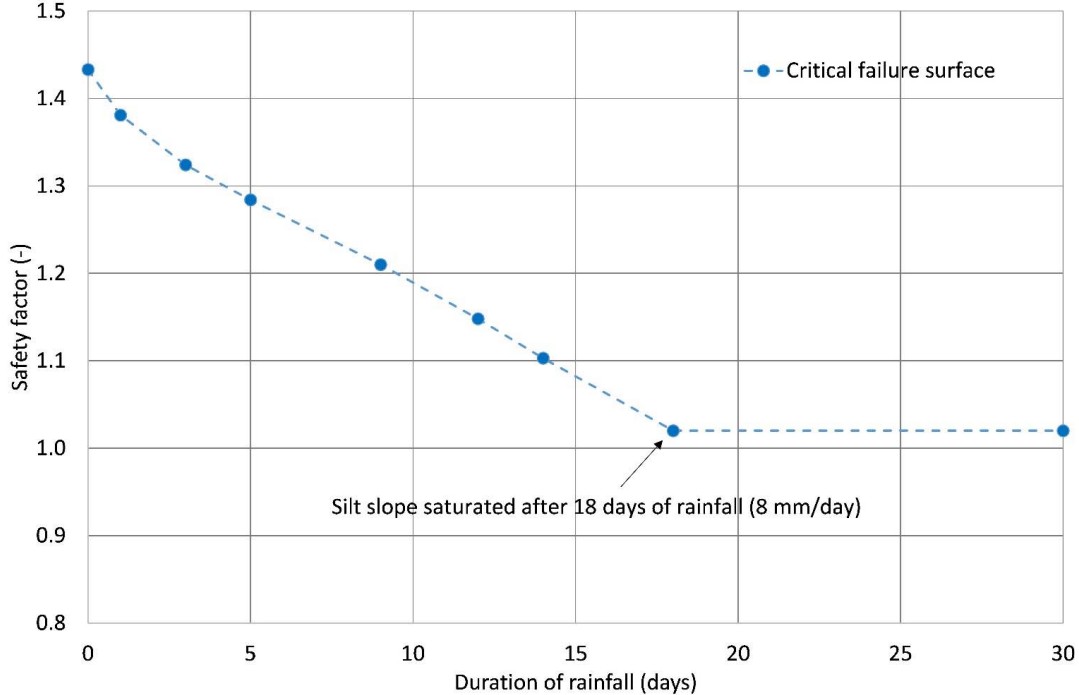

**Fig. 24 Development of slope safety factor for model slope (top layer of silt above clay) during extreme rainfall autumn year 2000 (240 mm in 30 days)**





**Table 1** Volume and weight parameters for shear box specimens ($\rho_s$=2.65 g/cm$^3$)

| Specimen ID | Depth m | $\gamma_d$ kN/m$^3$ | $n$ % | $e$ - | $w_{in\ situ}$ % | $S_{r,in\ situ}$ - | $\gamma_{in\ situ}$ kN/m$^3$ | $w_{sat}$ % | $\gamma_{sat}$ kN/m$^3$ |
|---|---|---|---|---|---|---|---|---|---|
| Silt S4 | 1.08 | 14.49 | 44.26 | 0.794 | 20.65 | 0.69 | 17.48 | 29.96 | 18.83 |
| Silt S5 | 1.12 | 14.49 | 44.26 | 0.794 | 21.06 | 0.70 | 17.54 | 29.97 | 18.83 |
| Silt S6 | 1.16 | 14.45 | 44.43 | 0.800 | 22.37 | 0.74 | 17.68 | 30.17 | 18.80 |
| Silt S7 | 1.20 | 14.27 | 45.09 | 0.821 | 29.06 | 0.94 | 18.42 | 30.99 | 18.69 |
| Silt S8 | 1.24 | 14.67 | 43.56 | 0.772 | 28.02 | 0.96 | 18.78 | 29.13 | 18.94 |


**Table 2** Summary of direct shear tests

| Test ID | Step No. | $u_{a,upper}$ kPa | $u_a$ kPa | $u_w$ kPa | $\sigma_{v,net} = \sigma_v - u_a$ [a] kPa | $u_a - u_w$ kPa | $\tau_{max}$ kPa |
|---|---|---|---|---|---|---|---|
| Silt S4 | 1 | 32.4 | 202.6 | 180.0 | 24.4 | 22.6 | 63.0 |
|  | 2 | 32.4 | 202.6 | 150.0 | 24.4 | 51.6 | 69.3 |
|  | 3 | 32.4 | 203.6 | 101.0 | 24.4 | 102.6 | 71.2 |
| Silt S5 | 1 | 49.6 | 102.3 | 89.0 | 44.0 | 13.3 | 76.0 |
|  | 2 | 49.6 | 102.3 | 81.0 | 44.0 | 21.3 | 80.7 |
|  | 3 | 49.6 | 102.3 | 60.0 | 44.0 | 42.3 | 81.9 |
|  | 4 | 49.6 | 102.3 | 21.0 | 44.0 | 81.3 | 82.4 |
| Silt S6 | 1 | 20.3 | 101.3 | 87.0 | 16.1 | 14.3 | 51.1 |
|  | 2 | 20.3 | 101.3 | 61.0 | 16.1 | 40.3 | 52.5 |
|  | 3 | 20.3 | 101.3 | 21.0 | 16.1 | 80.3 | 50.2 |
| Silt S7 | 1 | 21.3 | 0.0 | 0.0 | 20.2 | 0 | 38.1 |
|  | 2 | 20.3 | 0.0 | -6.4 | 19.3 | 6.4 | 53.3 |
|  | 3 | 20.3 | 0.0 | -9.3 | 19.3 | 9.3 | 62.3 |
|  | 4 | 26.3 | 202.6 | 180.0 | 18.7 | 22.6 | 69.4 |
|  | 5 | 26.3 | 202.6 | 160.0 | 18.7 | 42.6 | 70.7 |
|  | 6 | 26.3 | 202.6 | 120.0 | 18.7 | 82.6 | 70.2 |
| Silt S8 | 1 | 20.26 | 0 | 0 | 19.3 | 0.0 | 36.7 |
|  | 2 | 20.26 | 0 | -4.7 | 19.3 | 4.7 | 40.0 |
|  | 3 | 20.26 | 0 | -7.8 | 19.3 | 7.8 | 49.3 |
|  | 4 | 26.34 | 202.6 | 185.0 | 18.7 | 17.6 | 47.5 |
|  | 5 | 26.34 | 202.6 | 170.0 | 18.7 | 32.6 | 51.7 |
|  | 6 | 61.79 | 202.6 | 170.0 | 52.4 | 32.6 | 75.6 |
|  | 7 | 106.37 | 202.6 | 170.0 | 94.7 | 32.6 | 108.7 |

[a]Vertical net stress calculated from air pressures $u_{a,,upper}$ in upper test chamber (above load piston) and $u_a$ (in lower test chamber, containing shear box and specimen): $\sigma_{v,net}=0.9504u_{a,upper}-0.0314u_a$ (Ferrari, 2007)

