# Peer review of "Influence of extreme long-term rainfall and unsaturated soil properties on triggering of a landslide - a case study"

_Natural Hazards and Earth System Sciences, 2017_

## Referee Comment (RC1) · Anonymous Referee #1 · 8 Jan 2018

The manuscript deals with the interpretation of the failure of a slope in Norway occurred in 2000, taking into account the effect of soil suction on soil shear strength and its possible role in the slop failure evolution.

Such a topic is in line with the aims of NHESS. The English language is good and understandable.

However, the scientific content and the novelty of the manuscript are poor, while I believe that when a case study is presented for possible publication in a journal like NHESS, there must be novelty in the adopted modeling approach. Instead, as it will be detailed in the following specific comments, al the presented elaborations are based

on standard or even outdated approaches, so that the presented case study does not add any contribution to the understanding of triggering mechanisms in slopes like the one considered.

Therefore, I think the manuscript should be rejected.

Specific comments:

Section 2

The presented discussion of the effects of soil suction on effective shear strength in unsaturated conditions appears outdated (as are the models adopted afterwards to predict soil strength in the descirbed applications). Much work has been done in this topic. At least Lu and Likos (2006), Alonso et al. (2010), Lu et al. (2010), Nikoee et al. (2013), Greco and Gargano (2015) should be mentioned and maybe also applied, as the performance of the adopted models for the considered silt is extremely poor. Indeed, Greco and Gargano (2015) showed that silty soils are the ones for which the application of outdated approaches, like those adopted here for the evaluation of the contribution of soil suction to shear strength, leads to largely wrong results.

As the evaluation of the effects of unsaturated soil shear strength seems one of the main focuses of the manuscript, this is for me a very critical weak point.

Section 3.1

Page 5, lines 24-25. This is an example of the poor organization of the manuscript. The author touches here the possible effect on their results of the hysteresis in the water retention curve, without telling how he is willing to cope with it. Not even in section 4.3 ("Retention curve") he gives any information on this regard, and only in the sections devoteds to the case study, he comes out with other retention curves (and another soil, as well!), for which they modeled the hysteresis.

Section 4.1

Page 7, line 29. The numbering of tables should better follow their order of appearence in the text. Thus, Table 2 should be Table 1 and vice versa.

Section 4.2

Page 8, lines 3 and 5. The term "permeability", used here for the first time, and then repeated several times thorughout the manuscript, should be replaced with "hydraulic conductivity", which is the correct name of the variable the author refers to.

Page 8, lines 4 to 7. Another example of the lack of organization of the manuscript. Afetr referring only of silt samples (see Tables 1 and 2, as well as the entire sections 3.3 and 3.4), now sand suddenly comes out, and then also sandy silt and silty sand. The reader does not know anything about such soils.

Section 4.3

Page 8, line 12. Not only a graoh, but also the parmeters of the WRC according to Fredlund and Xing (1994) model should be provided. By the way, this curve is said to belong to a sandy silt.

Fig. 6. The expression proposed by Fredlund and Xing for the WRC relates suction with volumetric water content, whicle here the plot of gravimetric water cintent vs. suction is given. If a change of variables in the equation has been made (which is not totally correct, in my opinion, and not even necessary), this should be at least specified.

Section 5.3

Page 10, lines 15-20. It is not clear what the authors wants to point out. Is the water retention curve not representative of the tested soil? Or are the applied soil suction uncertain?

Section 5.6

It should be Section 5.5

Page 11, line 23. Other models, nowadays used at least as commonly as Vanapalli et al. (1996) take into account micromechanical effects.

Section 5.7

It should be Section 5.6

Page 11, lines 30-31. It is not true that the integranular forces exterted by suction act only along menisci, as they act also (maybe mainly, in most cases) as pressure exerted over the wteed portion of the external surface area of the solid particles (e.g. Greco and Gargano, 2015). See Lu and Likos (2006), which derive separately the contributions of menisci and pressure on the basis of thermodynamic considerations, and Nikoee et al. (2013), who propose a way to quantify them.

Page 12, line 7. The author seems to mix up two effects: the increased stiffness owing to overconsolidation does not imply an increased shear strength.

Section 6.1

Page 12, line 14. Now the simulations are carried out for two soils (silt and sand), after discussing only properties of silt. By the way, looking at the picture in fig. 2b, it appears that the soil profile at the considered site is layered, which may make things much different from the modeled cases. At least some discussion about this choice should be made.

Page 12, line 18. Probably this is a typo, but none of the three models presented in section 2.2 makes use of the product of water content times suction.

Page 12, lines 21-24, and figures 14 and 15. New hydraulic characteristic curves are here introduced, without any information about the "new" soils nor how the curves were estimated.

Page 12, lines 25-28. The "warm up" of the model with one year rainffall makes sense, but it has not only effects on the infiltration capacity (top surface boundary condition),

as said, but it is rather a way to start the simulation of the triggering rainfall event from realistic intial conditions.

Section 6.2.1

Page 13, line 12. I see the author's point, but I do not think it is correct to talk about a failure surface in a situation in which the minimum factor of safety is >2.

Sections 6.2.3 and 7

As a matter of fact, the author makes his slope stability analyses considering a contribution of suction to soil shear strength much smaller than measured (page 12, line 20). This has no effect on the results, as the slope needs to become fully saturated for the model to predict a failure along the slope.

So, I have serious doubts about the conclusion that "evaluation of slope stability based on unsaturated soil proerties will be increasingly important (...) to understanding rainfall triggering of landslides".

I even miss the overall message of the presented case study, as it does not seem so important to correctly evaluate the unsaturated shear strength of a soil along a slope where saturation is necessary to observe a failure.

Furthermore, even a substantial reduction of soil cohesion, compared to the experimentally determined values, must be introduced in order to obtain low values of the safety factor. A possible interpretation is that the actual layered soil profile deeply affects the infiltration process (e.g. Damiano et al., 2017) and, in turn, the slide triggering conditions. Thus, I wonder what we learn from the presented case study.

References

Alonso, E. E., J.-M. Pereira, J. Vaunat, and S. Olivella (2010), A microstructurally based effective stress for unsaturated soils, Geotechnique, 60(12), 913–925, doi:10.1680/geot.8.P.002.

[Figure]

Fredlund, D. G., and A. Xing (1994), Equations for the soil-water characteristic curve, Can. Geotech. J., 31, 521–532, doi:10.1139/t94-061.

E. Damiano, R. Greco, A. Guida, L. Olivares, L. Picarelli. Investigation on rainwater infiltration into layered shallow covers in pyroclastic soils and its effect on slope stability. Engineering Geology 220: 208-218, doi: 10.1016/j.enggeo.2017.02.006

R. Greco, R. Gargano. A novel equation for determining the suction stress of unsaturated soils from the water retention curve based on wetted surface area in pores. Water Resources Research. 51: 6143-6155, doi: 10.1002/2014WR016541

Lu, N., and W. J. Likos (2006), Suction stress characteristic curve for unsaturated soil, J. Geotech. Geoenviron. Eng., 132(2), 131–142, doi:10.1061/(ASCE)1090-0241(2006)132:2(131).

Lu, N., J. W. Godt, and D. T. Wu (2010), A closed-form equation for effective stress in unsaturated soil, Water Resour. Res., 46, W05515, doi:10.1029/2009WR008646.

Nikooee, E., G. Habibagahi, S. M. Hassanizadeh, and A. Ghahramani (2013), Effective stress in unsaturated soils: A thermodynamic approach based on the interfacial energy and hydromechanical coupling, Transp. Porous Media, 96(2), 369–396, doi:10.1007/s11242-012-0093-y.

---

## Referee Comment (RC2) · Anonymous Referee #2 · 12 Jan 2018

This manuscript describes the investigation and interpretation of a natural slope failure in eastern Norway.

The quality of English is good and the figures are clear. The content of the manuscript fits within the NHESS scope and is a topic of interest to an international audience.

The purpose and the aims of the paper are unclear to me and need to be more clearly defined. The manuscript describes the problem quite holistically. There are details of laboratory investigations, in situ observations and also modelling to back-analyse the failure. However, none of these aspects of the investigation are fully developed.

If the purpose of this paper is to outline different methods for slope stability interpreta-

tion, then I do not believe that this has been achieved. The methods used (laboratory and numerical modelling) are mainly routine and the novelty of the approaches is not demonstrated. Nor is there a particularly critical evaluation of their success or appropriateness when considering slopes in Norwegian soils.

If the purpose of the paper is to document and describe a case study for the academic community then I believe that further details of the site are required. Many of the details of the field investigation and many of the laboratory testing results are not included and instead the reader is directed to references. This suggests that much of the information that would constitute a case study is already published. The modelling aspects of the manuscript are not described in detail. As a reader I am not sure what the modelling proves or what new insight it provides into understanding slope failures in Norwegian soils.

The aims of the manuscript need to be more clearly outlined. I can see that the purpose is to 'fill in some gaps when it comes to unsaturated soil behaviour for Norwegian soils'. This needs to be clarified and more specific. The results also need to more clearly demonstrate this. At present, I believe that the manuscript is trying to cover many aspects of the case study and therefore presents a large number of different analyses and investigations but doesn't develop any new methods or interpretations. Nor does it provide enough detail for others to learn from, explore or build on the work. For example, I wonder if the slope stability analysis could be omitted.

I did not find a clear set of conclusions which relate directly to the described investigations. Nor did they give new insights into slope failures in Norway. I believe that there needs to be a clearer link between the aims, methods, results and conclusions presented in the manuscript. I have listed my comments relating to each of the conclusions below:

1) "Unsaturated shear tests. . .shear strength" [lines 5-10 on page 14]: The laboratory results are compared with prediction models. This sections states what happened and

how they compare (useful for the results section) but nothing is concluded. What does this show and why? 2) "Numerical seepage...compared to normal ground-water conditions" [ lines 12-15 on page 14]: The fact that rainfall can lead to elevated groundwater levels is a well-documented result and didn't require a numerical model to demonstrate this. Also, the model doesn't definitively prove that this was the cause of the landslide in 2000. What new behaviour did the unsaturated analysis show that wouldn't have been apparent in a saturated analysis? 3) "However, to reach critical slope stability...destabilize the slope" [lines 15-18 on page 14]: I'm not clear what this section is concluding. Please re-phrase to clarify. 4) "During the last decade....rainfall intensities and durations" [ lines 19-23 on page 14]: This seems like new information or summary/introductory information. It doesn't seem like a conclusion and I don't think that the work (as it is presented) indicates that an understanding of unsaturated soil properties will be increasingly important. This could be true, but I don' think that it is demonstrated in the current text.

In its current format and with the current focus, I believe that this manuscript should be rejected. This could be a useful and informative case study, but more detail and a clearer focus is required.

---

## Author Comment (AC1) · 11 Mar 2018

To both referees I want to express my thanks to both anonymous referees #1 and #2 for their efforts to give thorough and relevant comments and suggestions to the submitted paper. In my answers to the referee comments, I try to give answers to all comments. According to the procedure of the NHESS discussion process, a revised paper is not submitted at this time, which means that intended changes in the paper are only described principally in my answers. Some referee comments are general and not possible to answer/solve directly in the answers. At the start of my answer to each referee, I discuss these general comments, before answering the specific comments

from each of the Referees. For specific comments, I intend to improve the paper in line with my answers, or I give a clarification where there may be misunderstandings. Finally, I hope that my response to the referee comments will give me the opportunity to present a revised paper at a later stage. Referee #1 General comments I am happy that Referee #1 considers the topic of the paper within the aim of NHESS, although the main conclusions from that point on are not so positive. Some of the general comments might not be fully answered before a revised paper is submitted, but I would like to underline that I have considered each of the comments given by Referee #1 very seriously. I will not argue with the general view regarding to what extent a case study should include novelty, existing or new methods etc. However, I intend to include some ideas on interpretation of the unsaturated shear strength in the reviewed paper (not included in the first version of the paper). I can't be absolutely conclusive in my suggestions, since the laboratory tests were not designed to verify the method I will present. More work here could be interesting. The aims and scope of NHESS refers to "….a wide and diverse community of research scientists, practitioners, and decision makers….". In my opinion, this scope does not exclude case studies like mine. A practical case study may be useful for many of the groups listed above, by applying and combining aspects known from everyday practice, making the study within the reach of the reader (without being outdated). Still, I think it is true to say that the use of unsaturated soil mechanics is not so widely spread outside of academia, and NHESS is probably not the typical magazine for this special branch of geotechnics. That was also a reason for not diving too deep into testing methods and parameters of unsaturated soils. For many soils and countries there exists little unsaturated data. New data (of good quality) is therefore still an achievement and could give some credit in the scientific sense. Also coupling of local soil data with real/documented landslide events has scientific value, and maybe contributes to making the study interesting (in my view, that is…). Specific comments Section 2 Many thanks to Referee #1 for sharing references of relevant and recent literature. I will include more discussion of recent work in the reviewed paper, and consider whether some of them should be included for compari-

son with test data. "Outdated" is not synonymous with "old". Well-established (and old) methods are widely used in unsaturated research and practice, and may be attractive to the reader. E.g. the forever young WRC-model van Genuchten (1980), often preferred to the better (in my opinion) WRC-equation by Fredlund and Xing (1994), simple but comprehensible models for prediction of unsaturated shear strength by Vanapalli et al. (1996) and Öberg and Sällfors (1997); the even simpler bi-linear model by Fredlund et al. (1978), although other models as Khalili and Khabbaz (1998) and Lu and Likos (2006) are superior. Just one example: Alonso et al. (2010) actually discuss their model for evaluation of shear strength based on effective saturation rate by use of (among others) the simplest WRC-function of them all, from Brooks and Corey (1964). Section 3.1 Page 5, lines 24-25. I don't quite follow this criticism, as this section discusses ways to perform shear tests, by multi-stage or single-stage tests. Tests were run along a drying path for reasons discussed in this section. Thereby I did not see a reason to discuss how to deal with hysteresis here. In the infiltration calculations I do deal with this, by following the main wetting path of the WRC for the infiltration process. Section 4.1 Page 7, line 29. Numbering of Tables will be corrected. Section 4.2 Page 8, lines 3 and 5. The term "permeability" will be replaced with "hydraulic conductivity". Page 8, lines 4 to 7. The site description and presentation of soil data will be made more complete. References to sand and silt (abbreviated from "silty sand" and "sandy silt"), will be made consistent. Section 4.3 Page 8, line 12. Parameters of the WRC-equation (Fredlund and Xing, 1994) will be included. Fig. 6. The switch between water contents actually does not affect the equation or parameters in the Fredlund and Xing (1994) equation, as volumetric and gravimetric water content only differ by a constant for a given soil. I agree it is not necessary to do this switch, and will make this consistent in the reviewed paper (my apology is the geotechnician's habit of preferring the gravimetric water content). Section 5.3 Page 10, lines 15-20. This is a good question! The easy way out would be not to mention it (or even check it). There is however inherent uncertainty in actual water content in the specimen in the individual steps of a multi-stage shear test, as water content in the specimen may not be verified during the

test, only estimated from measurement of water flowing in/out of the specimen (which was done with a GDS pump, correcting for diffused air). I believe think deviating water content during tests (when comparing the WRC with applied suction) is not always reported; results are normally presented for applied suction without presenting this kind of uncertainty or mismatch in the data, but I have chosen to present it. As the silt layer was quite uniform (e.g. the void ratio varied within very narrow limits), and the determination of the WRC was quite thorough (Heyerdahl and Pabst, 2017), I have assumed that the measured WRC is representative. Concluding that water contents after tests are generally higher than expected from applied suction for the measured WRC, I need to make a choice for the interpretation. Correcting the suction values based on measured water content after the test actually makes the data collapse in a quite attractive manner. The answer to the question therefore is that applied (or "effective") suction is uncertain, and somewhat lower than the applied suction according to the controlling pressures (ua – uw) applied by axis translation. In a perfect laboratory test, these values would match. Section 5.6 The section number will be corrected. Page 11, line 23. I will include discussion of other interpretation methods for unsaturated shear strength in the theory chapter, and consider comparison with the test data. Section 5.7 Successive error... The section number will be corrected! Page 11, lines 30-31. My formulation is probably not good if it seems as if I generally mean that suction only acts in menisci at the grain contacts. That is of course not always the situation. What I wanted to express is that for low water contents the contacts at menisci will be important. I will also check with the suggested literature. Page 12, line 7 Here I do not agree. Experience generally shows that high stiffness caused by preconsolidation is related to higher shear strength than similar soils not exposed to preconsolidation. Reference is made to the classic work for OC clay (Ladd and Foott, 1974), and we know well that OC clay are stiffer than NC clay. The same goes for other soil types when compressed and then unloaded. The question still is whether suction alone causes a similar effect, not only on the volumetric response. Maybe at the moment only a hypothesis – but not a mix-up with stiffness. Section 6.1 Page 12, line 14. The site is definitely layered. To

account for all the layers would be virtually impossible, and the question is whether a detailed layer modelling is possible, when considering uncertainty in lateral continuity and layer thickness. The laboratory data is assumed representative for the variety in the soil mass. The soils are not completely different, mainly consisting of sandy silt and silty sand, and the hydraulic conductivity of these layers only varies with about one order of magnitude. I will attempt to improve the discussion of the numerical modelling. The idea was to avoid numerical trouble with rather arbitrary layers of almost equal soils, for which the variation in soil parameters is not well-defined. Instead, the effect of assuming more or less homogeneous conditions for these soil types was checked. Page 12, line 18. I wanted to be conservative and not utilise the full strength of the silt as measured, meaning that water content was for simplicity assumed to represent $\chi$. At least such values are documented from the tests, both for sandy silt and for sand (Heyerdahl, 2016). It was also a choice available in the calculation software (Geo-Slope_International, 2015). I will consider including a more precise strength function for the revised analyses. Page 12, line 21-24, Figures 14 and 15. The soils will be presented more clearly, with information about how the hydraulic curves were estimated based on the WRC-functions and measured saturated hydraulic permeability. Page 12, line 25-28. I do agree. However, the start of line 25 points not to the infiltration capacity, but to the applied rainfall at the top boundary. Section 6.2.1 Page 13, line 12. Agreed, I could rather use the term "critical shear surface"? Section 6.2.3 and 7 The numerical analyses (unfortuneately..) show that unsaturated shear strength is not critical for landslide release here, since the slope saturates by groundwater rise from the clay, and due to the high measured cohesion. Unsaturated flow is however the guiding mechanism that destabilises the slope, and is connected to the title: "..extreme long-term rainfall and unsaturated soil properties...". In this way, I don't think the title is misleading. I wish to present the unsaturated shear strength data in the paper, as I consider them a step forward for unsaturated research on Norwegian silty soils (whatever that may be worth). One way to make the unsaturated strength more relevant to the study, would be to apply the data to triggering of shallow surfaces, exposing the slopes to more intense

short-term rainfall, which also occurs in these types of soils. (Again; it will be difficult to trigger shallow landslides for the high measured cohesion, which means that such an analysis will have to lean on a variation in this parameter).

Alonso, E. E., Pereira, J.-M., Vaunat, J., and Olivella, S., 2010, A microstructurally based effective stress for unsaturated soils: Gèotechnique, v. 60, no. 12, p. 913-925. Brooks, R. H., and Corey, A. T., 1964, Hydraulic properties of porous media., Colorado State University, Ft.Collins, Colo. Hydrology paper No.3, (March). Fredlund, D. G., Morgenstern, N. R., and Widger, R. A., 1978, The shear strength of unsaturated soils: Canadian Geotechnical Journal, v. 15, no. 3, p. 313-321. Fredlund, D. G., and Xing, A., 1994, Equations for the soil-water characteristic curve: Canadian Geotechnical Journal, v. 31, p. 521-532. Geo-Slope_International, 2015, GeoStudio 2012, in GEO-SLOPE_International, L., ed., Volume August 2015 release Calgary, Alberta, Canada. Heyerdahl, H., 2016, Rainfall-induced landslides in Quaternary soils in Norway, in Proceedings 3rd European Conference on Unsaturated Soils (E-UNSAT 2016), Paris, France, 12-14 September 2016. Heyerdahl, H., and Pabst, T., 2017, Comparison of experimental and predictive approaches for determination of water retention curves of intact samples of Quaternary soils Journal of Geotechnical and Geological engineering, p. 1-21. Khalili, N., and Khabbaz, M. H., 1998, A unique relationship for X for the determination of the shear strength of unsaturated soils: Géotechnique, v. 48, no. 5, p. 7. Ladd, C. C., and Foott, R., 1974, New design procedure for stability of soft clays: Journal of Geotechnical Engineering, ASCE, v. 100, no. GT7, p. 24. Lu, N., and Likos, W. J., 2006, Suction stress characteristic curve for unsaturated soil: Journal of Geotechnical and Geoenvironmental Engineering, v. 132, no. 2, p. 131-142. van Genuchten, M. T., 1980, A Closed-form Equation for Predicting the Hydraulic Conductivity of Unsaturated Soils, SoilSci.Soc.Am.J., Volume 44, p. 892-898. Vanapalli, S. K., Fredlund, D. G., Pufahl, D. E., and Clifton, A. W., 1996, Model for the prediction of shear strength with respect to soil suction: Canadian Geotechnical Journal, v. 33, no. 3, p. 379-392. Öberg, A. L., and Sällfors, G., 1997, Determination of shear strength parameters of unsaturated silts and sands based on the water retention curve: Geotechnical

Testing Journal, v. 20, no. 1, p. 40-48.

---

## Author Comment (AC2) · 11 Mar 2018

To both referees I want to express my thanks to both anonymous referees #1 and #2 for their efforts to give thorough and relevant comments and suggestions to the submitted paper. In my answers to the referee comments, I try to give answers to all comments. According to the procedure of the NHESS discussion process, a revised paper is not submitted at this time, which means that intended changes in the paper are only described principally in my answers. Some referee comments are general and not possible to answer/solve directly in the answers. At the start of my answer to each referee, I discuss these general comments, before answering the specific comments

from each of the Referees. For specific comments, I intend to improve the paper in line with my answers, or I give a clarification where there may be misunderstandings. Finally, I hope that my response to the referee comments will give me the opportunity to present a revised paper at a later stage. Referee #2 I am happy that Referee #2 considers the topic appropriate for NHESS, and I thank the referee for many constructive suggestions and comments. I have given them all a lot of thought, and hope my revised explanations and answers makes the purpose and aims of the work more clear, and that my answers to referee comments generally are found satisfactory. The main motivation and purpose of the study is to improve the general understanding of reasons for increased landslide activity as experienced in Norway in recent years, through a case study based on events in year 2000. Applying internationally recognized methods for unsaturated slope analysis is clearly a part of this motivation, including collection of unsaturated parameters for soils not investigated before. Such methods may be considered "routine" (although rarely found outside academia), but data is new and to some extent surprising, as the unsaturated shear strength of silt. As a novelty I will now also include a suggestion for estimation of unsaturated shear strength in soils previously exposed to high suction (left out in the first version). For the case study I also think that the fact that soil parameters come from tests on intact specimens makes them more valuable, particularly for granular soils, that are normally hard to sample and test in an intact state. To further justify the study: Many landslide studies have been presented internationally with virtually no real soil data. This case study covers many elements often missing in such studies: Site specific shear strength for saturated and unsaturated conditions, permeability and water retention properties and ground investigations, in addition to coupling to a documented landslide event and rainfall records. Regarding "fill in some gaps. . .": Elaboration on this will be included in section 1.1, mainly referring to the scarcity, i.e. almost total absence, of local data ("fill in some gaps" is actually an understatement). I recognize that a much more complete description of the site is necessary. I should not expect the referees to be clairvoyant! More details will be included to improve and complete the description of the site. Some

details of the testing will still be found in referenced literature (e.g. details of WRC-testing (Heyerdahl and Pabst, 2017) and shear testing of sand (Heyerdahl, 2016), but main results will be summarized. Regarding the conclusions: Considering results from numerical analyses, infiltration through unsaturated soil layers is the most important process. Failure indeed occurs primarily in saturated soil layers; however, this is a result of the parameters found. Starting the work it was not obvious what the outcome would be. For understanding the development of critical situations, it would be hard to omit slope stability analyses completely, although infiltration is the governing factor. A point of learning is the fact that even with considerable data collection and soil testing, a "perfect explanation" for landslide triggering of a landslide was not reached (although the conceptual explanation is very clear). Data from the study could be applied to study infiltration and slope failure of other slopes in Norway in similar soils, for instance shallow landslides resulting from infiltration on in "infinite slopes"; this would however go beyond the scope of this particular paper. For such studies, this case study show that cohesion of intact specimens is an important factor and govern the resulting slope stability. Answers to specific comments: 1) Lines 5-10 on page 14 To improve the conclusions, comments will be included regarding why results are as they are (at least what I assume may be the reasons). 2) Lines 12-15 on page 14 No doubt, processes of infiltration and groundwater rise are generally known. However, in my view, a documented case study based on considerable data should have some interest. Not the least, it should be interesting what amount of rainfall that is necessary to induce a critical situation. A case study of an actual landslide should be an appropriate format for such investigations. The numerical analysis does not prove everything – but combined with the observed failure geometry, things start to fall in place. Regarding the question "What new behaviour did the unsaturated analysis show that wouldn't have been apparent in a saturated analysis": The analysis applies site specific soil data and actual rainfall records to see how the response of the initially unsaturated slope matches with observed failure. The saturated analysis would give a different result for an infiltration analysis. It is recognized that the slope failure occurs in a saturated slope. This was

not possible to know at the starting point of the study. As mentioned, unsaturated shear strength of Norwegian soils, performed on intact specimens, are unprecedented and should give some value. Also, the results showing 3) Lines 15-18 on page 14 The section will be improved. The analyses show that even for a saturated slope, failure does not occur. Effective friction angles for silty sand and sandy silt are quite reliable, which puts the uncertainty mostly on the cohesion. The cohesion must therefore be smaller for the gross volume of the soil than measured in specimens tested. 4) I will moved the phrase "During the last decade....rainfall intensities and durations" to the introduction part, adding to the motivation for the study. In the conclusion, I will elaborate more on the importance of increased knowledge on unsaturated infiltration (and shear strength), as the unsaturated state must be the starting point for triggering of a rainfall triggered landslide. Heyerdahl, H., 2016, Rainfall-induced landslides in Quaternary soils in Norway, in Proceedings 3rd European Conference on Unsaturated Soils (E-UNSAT 2016), Paris, France, 12-14 September 2016. Heyerdahl, H., and Pabst, T., 2017, Comparison of experimental and predictive approaches for determination of water retention curves of intact samples of Quaternary soils Journal of Geotechnical and Geological engineering, p. 1-21.